# Learning Truncated Causal History Model for Video Restoration

♣**Amirhosein Ghasemabadi**
ECE Department, University of Alberta
ghasemab@ualberta.ca

♣**Muhammad Kamran Janjua**
Huawei Technologies, Canada
kamran.janjua@huawei.com

**Mohammad Salameh**
Huawei Technologies, Canada
mohammad.salameh@huawei.com

**Di Niu**
ECE Department, University of Alberta
dniu@ualberta.ca

## Abstract

One key challenge to video restoration is to model the transition dynamics of video frames governed by motion. In this work, we propose TURTLE to learn the **TRU**nca**T**ed causa**L** history mod**E**l for efficient and high-performing video restoration. Unlike traditional methods that process a range of contextual frames in parallel, Turtle enhances efficiency by storing and summarizing a truncated history of the input frame latent representation into an evolving historical state. This is achieved through a sophisticated similarity-based retrieval mechanism that implicitly accounts for inter-frame motion and alignment. The causal design in TURTLE enables recurrence in inference through state-memorized historical features while allowing parallel training by sampling truncated video clips. We report new state-of-the-art results on a multitude of video restoration benchmark tasks, including video desnowing, nighttime video deraining, video raindrops and rain streak removal, video super-resolution, real-world and synthetic video deblurring, and blind video denoising while reducing the computational cost compared to existing best contextual methods on all these tasks.

○ https://kjanjua26.github.io/turtle/

## 1 Introduction

Video restoration aims to restore degraded low-quality videos. Degradation in videos occurs due to noise during the acquisition process, camera sensor faults, or external factors such as weather or motion blur [53, 38]. Several methods in the literature process the entire video either in parallel or with recurrence in design. In the former case, multiple contextual frames are processed simultaneously to facilitate information fusion and flow, which leads to increased memory consumption and inference cost as the context size increases [63, 4, 69, 86, 58, 28, 26, 5, 34, 62]. Methods with recurrence in design reuse the same network to process new frame sequentially based on previously refined ones [54, 14, 21, 25, 6, 7, 42, 57]. Such sequential processing approaches often result in cumulative errors, leading to information loss in long-range temporal dependency modeling [8] and limiting parallelization capabilities.

Recently, methods based on state space models (SSMs) have seen applications across several machine vision tasks, including image restoration [19, 56], and video understanding [30]. While Video-Mamba [30] proposes a state space model for video understanding, the learned state space does not reason at the pixel level and, hence, can suffer from information collapse in restoration tasks [77].

---

♣ indicates equal contribution.

38th Conference on Neural Information Processing Systems (NeurIPS 2024).

Additionally, the state evolves over time with respect to motion that affects the entire trajectory non-uniformly [51] at the pixel level. Therefore, it is pertinent to learn a model capable of summarizing the history[1] of the input as it operates on the spatiotemporal structure of the input video.

In this work, we present "TURTLE", a new video restoration framework to learn the **TR**unca**T**ed causa**L** history mod**E**l of a video. TURTLE employs the proposed Causal History Model (CHM) to align and borrow information from previously processed frames, maximizing feature utilization and efficiency by leveraging the frame history to enhance restoration quality. We outline our contributions.

- TURTLE's encoder processes each frame individually, while its decoder, based on the proposed Causal History Model (CHM), reuses features from previously restored frames. This structure dynamically propagates features and compensates for lost or obscured information by conditioning the decoder on the frame history. CHM models the evolving state and compensates the history for motion relative to the input. Further, it learns to control the effect of history frames by scoring and aggregating motion-compensated features according to their relevance to the restoration of the current frame.

- TURTLE facilitates training parallelism by sampling short clips from the entire video sequence. In inference, TURTLE's recurrent view implicitly maintains the entire trajectory ensuring effective frame restoration.

- TURTLE sets new state-of-the-art results on several benchmark datasets and video restoration tasks, including video desnowing, nighttime video deraining, video raindrops and rain streak removal, video super-resolution, real and synthetic video deblurring, and achieves competitive results on the blind video denoising task.

## 2 Related Work

Video restoration is studied from several facets, mainly distributed in how the motion is estimated and compensated for in the learning procedure and how the frames are processed. Additional literature review is deferred to appendix G.

**Motion Compensation in Video Restoration.** Motion estimation and compensation are crucial for correcting camera and object movements in video restoration. Several methods employ optical flow to explicitly estimate motion and devise a compensation strategy as part of the learning procedure, such as deformable convolutions [33, 34], or flow refinement [23]. However, optical flow can struggle with degraded inputs [84, 3, 20], often requiring several refinement stages to achieve precise flow estimation. On the other end, methods also rely on the implicit learning of correspondences in the latent space across the temporal resolution of the video; a few techniques include temporal shift modules [29], non-local search [64, 32, 85], or deformable convolutions [69, 13, 80].

**Video Processing Methods.** There is a similar distinction in how a video is processed, with several methods opting for either recurrence in design or restoring several frames simultaneously. Parallel methods, also known as sliding window methods, process multiple frames simultaneously. This sliding window approach can lead to inefficiencies in feature utilization and increased computational costs [63, 4, 69, 86, 58, 28, 26, 5, 34, 62, 9]. Although effective in learning joint features from the entire input context, their size and computational demands often render them unsuitable for resource-constrained devices. Conversely, recurrent methods restore frames sequentially, using multiple stages to propagate latent features [87, 81, 82]. These methods are prone to information loss [33]. Furthermore, while typical video restoration methods in the literature often rely on context from both past and future neighboring frames [34, 33, 29], TURTLE is causal in design, focuses on using only past frames. This approach allows TURTLE to apply in scenarios like streaming and online video restoration, where future frames are unavailable.

## 3 Methodology

Consider a low-quality video $\mathbf{I}^{\mathrm{LQ}} \in \mathbb{R}^{T \times H \times W \times C}$, where $T, H, W, C$ denote the temporal resolution, height, width, and number of channels, respectively, that has been degraded with some degradation

---

[1]In this manuscript, the term "history" refers to temporally previous frames with respect to the input frame.

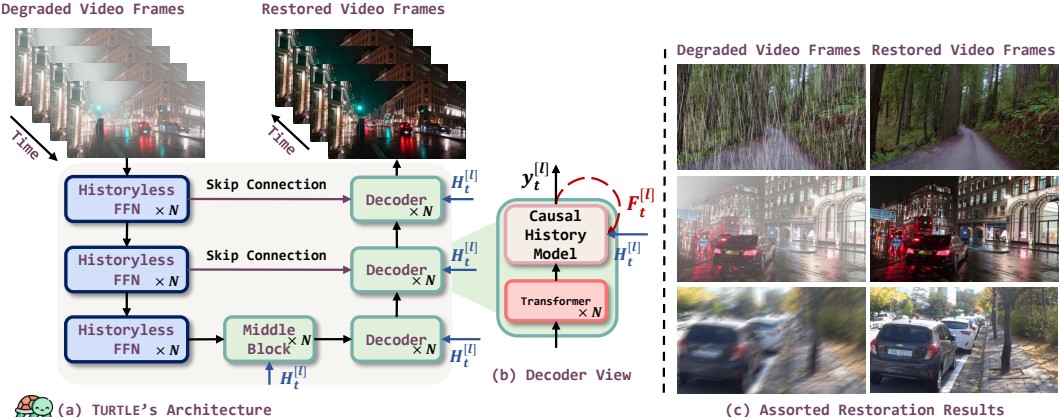

Figure 1: **TURTLE's Architecture.** The overall architecture diagram of the proposed method. TURTLE is a U-Net [52] style architecture, wherein the encoder blocks are historyless feedforward blocks, while the decoder couples the causal history model (CHM) to condition the restoration procedure on truncated history of the input. We also present assorted restoration examples on the right–frame taken from video raindrops and rain streak removal [71], night deraining [47], and video deblurring [41] tasks, respectively.

$d \in \mathbb{D}$. The goal of video restoration is to learn a model $M_\theta$ parameterized by $\theta$ to restore high-quality video $\mathbf{I}^{\text{HQ}} \in \mathbb{R}^{T \times sH \times sW \times C}$, where $s$ is the scale factor (where $s > 1$ for video super-resolution). To this end, we propose TURTLE, a U-Net style [52] architecture, to process, and restore a single frame at any given timestep conditioned on the truncated history of the given frame. TURTLE's encoder focuses only on a single frame input and does not consider the broader temporal context of the video sequence. In contrast, the decoder, however, utilizes features from previously restored frames. This setup facilitates a dynamic propagation of features through time, effectively compensating for information that may be lost or obscured in the input frame. More specifically, we condition a decoder block at the different U-Net stages on the history of the frames. Given a frame at timestep $t$, each block learns to model the causal relationship $p(\mathbf{y}_t | \mathbf{F}_t, \mathbf{H}_t)$, where $\mathbf{y}_t$ is the output of a decoder block, $\mathbf{F}_t$ is the input feature map of the decoder block, and $\mathbf{H}_t$ is the history of corresponding features maps from the previous frames at the same block. We train the architecture with the standard $\text{L}_1$ loss function: $\mathcal{L} = \frac{1}{N} \sum_{i=1}^{N} \|\mathbf{I}^{\text{GT}} - \mathbf{I}^{\text{HQ}}\|_1$ for all the restoration tasks. We present the visual illustration of TURTLE's architecture in Figure 1.

## 3.1 Architecture Design

Given a model $M_\theta$, let $\mathbf{F}_t^{[l]}$ denote the feature map of a frame at timestep $t$, taken from $M_\theta$ at layer $l$. We, then, utilize $\mathbf{F}_t^{[l]}$ to construct the causal history states denoted as $\mathbf{H}_t^{[l]} \in \mathbb{R}^{\tau \times h^l \times w^l \times c^l}$, where $\tau$ is the truncation factor (or length of the history), $h, w$ denote spatial resolution of the history, and $c$ denotes the channels. More specifically, $\mathbf{H}_t^{[l]} = \{\mathbf{F}_{t-\tau}^{[l]} \oplus \mathbf{F}_{t-\tau+1}^{[l]} \oplus \ldots \oplus \mathbf{F}_{t-1}^{[l]}\} \in \mathbb{R}^{\tau \times h^l \times w^l \times c^l}$, where $\oplus$ is the concatenation operation. We denote the motion-compensated history at timestep $t$ as $\hat{\mathbf{H}}_t^{[l]}$, which is compensated for motion with respect to the input frame features $\mathbf{F}_t^{[l]}$. In this work, the state refers to the representation of a frame of the video. Further, history states (or causal history states) refers to a set of certain frame features previous to the input at some timestep.

TURTLE's encoder learns a representation of each frame by downsampling the spatial resolution, while inflating the channel dimensions by a factor of 2. At each stage of the encoder, we opt for several stacked convolutional feedforward blocks, termed as Historyless FFN,[2] The learned representation at the last encoder stage onwards is fed to a running history queue $\mathcal{Q}$ of length $\gamma$.[3] We empirically set $\gamma = 5$ for all the tasks, and consider sequence of 5 frames. The entire video sequence is reshaped

---

[2]We use the term "Historyless" because the encoder is not conditioned on the history of input.

[3]The space complexity is limited to $\mathcal{O}(\gamma \times h \times w \times c)$ and $(\gamma \times h \times w \times c) << (T \times H \times W \times C)$ because $\gamma << T$, and $h << H, w << W$, and all enqueue and dequeue operations are $\mathcal{O}(1)$ in time complexity.

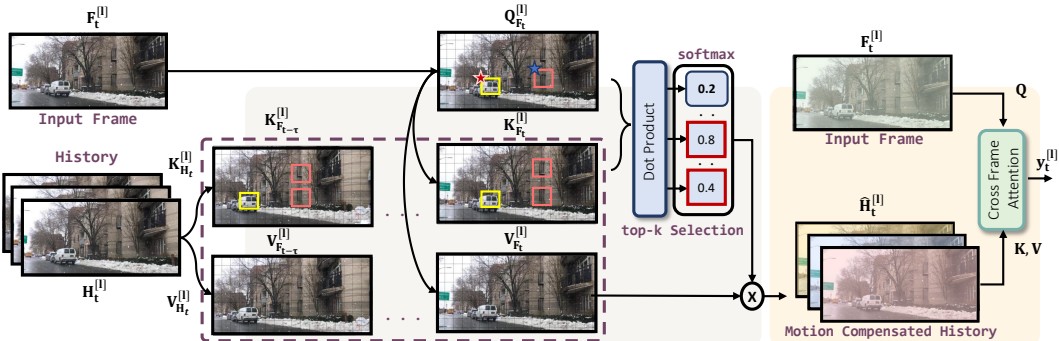

Figure 2: **Causal History Model.** The diagrammatic illustration of the proposed Causal History Model (CHM) detailing the internal function. In the initial phase, for each patch in the current frame (denoted by the stars), we identify and implicitly align the *top-k* similar patches in the history. In the subsequent phase, we score and aggregate features from this aligned history to create a refined output that blends the input frame features with pertinent history data. We visualize frames in this diagram for exposition, but in practice the procedure operates on the feature maps.

into $\mathbb{R}^{\frac{T}{\gamma} \times H \times W \times C}$ thereby allowing parallelism in training while maintaining a dense representation of history states to condition the reconstruction procedure on.

The decoder takes the feature map of the current frame, $\mathbf{F}_t^{[l]}$, and the history states $\mathbf{H}_t^{[l]}$. We propose a motion compensation module that operates on the feature space to implicitly align history states with respect to the input frame. Next, a dynamic router learns to control the effect of history frames by scoring and aggregating motion-compensated features based on their relevance to the restoration of the current frame. Such a procedure accentuates the aligned history such that the following stages of the decoder can learn to reconstruct the high-quality frame appropriately. Both of these procedures combine to form the Causal History Model CHM($\mathbf{F}_t^{[l]}, \mathbf{H}_t^{[l]}$), detailed in section 3.2. Further, multiple CHMs are stacked as black box layers at different stages to construct the decoder of TURTLE.

## 3.2 Causal History Model

CHM learns to align the history states with respect to the input feature map. Further, there could still exist potential degradation differences at the same feature locations along the entire sequence in the motion-compensated history states. To this end, CHM re-weights the sequence along the temporal dimension to accentuate significant features and suppress irrelevant ones. Let $\hat{\mathbf{H}}_t^{[l]} \in \mathbb{R}^{(\tau+1) \times h^l \times w^l \times c^l}$ denote the motion-compensated causal history states, and let input feature map be $\mathbf{F}_t^{[l]} \in \mathbb{R}^{h^l \times w^l \times c^l}$. Let the transformation on the history states to align the features be denoted by $\phi_t$, and let $\psi_t$ denote the re-weighting scheme. If the output is given by $\mathbf{y}_t^{[l]} \in \mathbb{R}^{h^l \times w^l \times c^l}$, we then, formalize the Causal History Model (CHM) as,

$$\hat{\mathbf{H}}_t^{[l]} = \phi_t(\mathbf{H}_t^{[l]}, \mathbf{F}_t^{[l]}) \oplus \mathcal{B}_t(\mathbf{F}_t^{[l]}), \tag{1}$$

$$\mathbf{y}_t^{[l]} = \psi_t(\hat{\mathbf{H}}_t^{[l]}, \mathbf{F}_t^{[l]}) + \mathcal{D}_t(\mathbf{F}_t^{[l]}). \tag{2}$$

In eq. (1), $\mathcal{B}_t$ denotes transformation on the input, and $\mathcal{D}$ denotes the skip connection, while $\oplus$ is the concatenation operation. In practice, we learn $\phi_t$, and the input transformation matrix $\mathcal{B}$ following the procedure described in *State Align Block*, while $\psi_t$ is detailed in *Frame History Router*. We present a visual illustration of Causal History Model (CHM) in fig. 2. We also present a special case of (CHM) in appendix D, wherein we consider optimally compensated motion in videos.

**State Align Block ($\phi$).** State Align Block ($\phi$) implicitly tracks and aligns the corresponding regions defined as groups of pixels (or patches) ($p_1 \times p_2$)—across each frame in the history. State Align Block computes attention scores through a dot product between any given patch from the current frame and

all the patches from the history. Given the input feature map of a frame $\mathbf{F}_t^{[l]} \in \mathbb{R}^{h^l \times w^l \times c^l}$, we calculate the patched projections as, $\mathbf{Q}_{\mathbf{F}_t}^{[l]}, \mathbf{K}_{\mathbf{F}_t}^{[l]}, \mathbf{V}_{\mathbf{F}_t}^{[l]} \in \mathbb{R}^{\frac{h^l}{p_1} \times \frac{w^l}{p_2} \times (cp_1p_2)^l}$, i.e., $\mathbf{Q}_{\mathbf{F}_t}^{[l]}, \mathbf{K}_{\mathbf{F}_t}^{[l]}, \mathbf{V}_{\mathbf{F}_t}^{[l]} \leftarrow \mathbf{F}_t^{[l]} W^{\mathbf{F}_t^{[l]}}$

where $W^{\mathbf{F}_t^{[l]}}$ is a learnable parameter matrix. For exposition, let the dimensions of projections be $\mathbb{R}^{n_h^l \times n_w^l \times d^l}$, we subsequently rearrange the patches to $\mathbb{R}^{(n_h n_w)^l \times d^l}$. Here, $n_h^l = \frac{h^l}{p_1}$, and $n_w^l = \frac{w^l}{p_2}$ denote the number of patches along the height and width dimension, and $d^l = (cp_1p_2)^l$ represents the dimension of each patch. Formally, we define the history states $\mathbf{H}_t^{[l]}$ as a set of keys and values to facilitate the attention mechanism as,

$$\mathbf{H}_t^{[l]} = \{\mathbf{K}_{\mathbf{H}_t}^{[l]}, \mathbf{V}_{\mathbf{H}_t}^{[l]}\}, \tag{3}$$

where $\mathbf{K}_{\mathbf{H}_t}^{[l]}$, and $\mathbf{V}_{\mathbf{H}_t}^{[l]}$ are formally written as $\mathbf{K}_{\mathbf{H}_t}^{[l]} = \{\mathbf{K}_{\mathbf{F}_{t-\tau}}^{[l]}, \mathbf{K}_{\mathbf{F}_{t-\tau+1}}^{[l]}, \ldots, \mathbf{K}_{\mathbf{F}_{t-1}}^{[l]}\} \in \mathbb{R}^{\tau \times n_h n_w \times d}$, and $\mathbf{V}_{\mathbf{H}_t}^{[l]} = \{\mathbf{V}_{\mathbf{F}_{t-\tau}}^{[l]}, \mathbf{V}_{\mathbf{F}_{t-\tau+1}}^{[l]}, \ldots, \mathbf{V}_{\mathbf{F}_{t-1}}^{[l]}\} \in \mathbb{R}^{\tau \times n_h n_w \times d}$.

We, then, compute the attention, and limit it to the *top-k* most similar patches in the key vector for each patch in the query vector, and, hence, focus solely on those that align closely. This prevents the inclusion of unrelated patches, which can, potentially, introduce irrelevant correlations, and obscure principal features. We, then, formalize the *top-k* selection procedure as,

$$\mathbf{A}_t^{[l]} = (\mathbf{Q}_{\mathbf{F}_t}^{[l]} \cdot \mathbf{K}_{\mathbf{H}_t}^{[l]})/\alpha \in \mathbb{R}^{\tau \times (n_h^l n_w^l) \times (n_h^l n_w^l)}, \tag{4}$$

$$\mathbf{A}^{*[l]}_t = \begin{cases} x, & \text{if } x \in \text{topk}_{i \in (n_h^l n_w^l)}(\mathbf{A}_{(:,:,\mathbf{i})}^{[l]})_t, k), \\ -\infty, & \text{otherwise} \end{cases} \tag{5}$$

where $\alpha$ is a learnable parameter to scale the dot product, and $\mathbf{A}_{(:,:,\mathbf{i})}$ denotes the $i^{\text{th}}$ patch along the second dimension. $\mathbf{A}^{*[l]}_t$ masks the non *top-k* scores, and replaces with $-\infty$ to allow for softmax computation. In other words, each patch is compensated for with respect to its *top-k* similar, and salient patches across the trajectory. Such a procedure allows for soft alignment, and encourages each patch to borrow information from its most similar temporal neighbors, i.e., a one-to-*top-k* temporal correspondence is learned. Given the *top-k* scores, we compute the motion-compensated history states $\hat{\mathbf{H}}_t^{[l]}$ as follows,

$$\hat{\mathbf{H}}_t^{[l]} = \left[\sigma(\mathbf{A}^{*[l]}_t)\mathbf{V}_{\mathbf{H}_t}^{[l]}\right] W^{\hat{\mathbf{H}}_t^{[l]}} \oplus \mathcal{B}_t(\mathbf{F}_t^{[l]}), \tag{6}$$

where $\sigma$ is the softmax operator, $\oplus$ is the concatenation operator, $W^{\hat{\mathbf{H}}_t^{[l]}}$ is the parameter matrix learned with gradient descent, and $\mathcal{B}$ is a transformation on the input $\mathbf{F}_t^{[l]}$ realized through self-attention along the spatial dimensions [39, 66]. In eq. (6), $\phi_t(\mathbf{H}_t^{[l]}, \mathbf{F}_t^{[l]}) = \left[\sigma(\mathbf{A}^{*[l]}_t)\mathbf{V}_{\mathbf{H}_t}^{[l]}\right] W^{\hat{\mathbf{H}}_t^{[l]}}$ which follows from eq. (1).

**Frame History Router** ($\psi$). Given the motion-compensated history states $\hat{\mathbf{H}}_t^{[l]} \in \mathbb{R}^{(\tau+1) \times h^l \times w^l \times c^l}$ and the input features $\mathbf{F}_t^{[l]} \in \mathbb{R}^{h^l \times w^l \times c^l}$, Frame History Router ($\psi$) learns to route and aggregate critical features for the restoration of the input frame. To this end, we compute the query vector from $\mathbf{F}_t^{[l]}$ through the transformation matrix $W^{\mathbf{Q}_t^{[l]}}$, resulting in $\mathbf{Q}_t^{[l]} \leftarrow \mathbf{F}_t^{[l]} W^{\mathbf{Q}_t^{[l]}}$. Similarly, the key and value vectors are derived from $\hat{\mathbf{H}}_t^{[l]}$, and are parameterized $W^{\hat{\mathbf{H}}_t^{[l]}}$, i.e., $\mathbf{K}_t^{[l]}, \mathbf{V}_t^{[l]} \leftarrow \hat{\mathbf{H}}_t^{[l]} W^{\hat{\mathbf{H}}_t^{[l]}}$.

This configuration enables cross-frame channel attention, where the query from $\mathbf{F}_t^{[l]}$ attends to channels from both $\hat{\mathbf{H}}_t^{[l]}$ and $\mathbf{F}_t^{[l]}$, and accentuates temporal history states as necessary in order to restore the given frame. The cross-channel attention map $\mathbf{A} \in \mathbb{R}^{(\tau+1)c^l \times c^l}$ is then computed through the dot product, i.e., $\mathbf{A}_t^{[l]} = (\mathbf{Q}_t^{[l]} \cdot \mathbf{K}_t^{[l]})/\alpha \in \mathbb{R}^{(\tau+1)c^l \times c^l}$, where $\alpha$ is the scale factor to control the dot product magnitude. Note that, we overload the notation $\mathbf{A}_t^{[l]}$ for exposition. The output feature map, $\mathbf{y}_t^{[l]}$ takes the shape $\mathbb{R}^{h^l \times w^l \times c^l}$ since the attention matrix takes the shape $\in \mathbb{R}^{(\tau+1)c^l \times c^l}$, while $\mathbf{V}_t^{[l]}$ is $\in \mathbb{R}^{(\tau+1)c^l \times h^l \times w^l}$.[4] If $\sigma$ denotes the softmax operator, and $\mathcal{D}$ is the skip connection, we then

---

[4]This is because in practice, channels are collapsed in the temporal dimension.

Table 1: **Night Video Deraining Results.**

| Method | PSNR↑ | SSIM↑ |
|---|---|---|
| FDM [22] | 23.49 | 0.7657 |
| DSTFM [46] | 17.82 | 0.6486 |
| WeatherDiff [43] | 20.98 | 0.6697 |
| RMFD [75] | 16.18 | 0.6402 |
| DLF [74] | 15.17 | 0.6307 |
| HRIR [31] | 16.83 | 0.6481 |
| MetaRain (Meta) [47] | 23.49 | 0.7171 |
| MetaRain (Scrt) [47] | 22.21 | 0.6723 |
| NightRain [35] | 26.73 | 0.8647 |
| TURTLE | **29.26** | **0.9250** |

Table 2: **Video Desnowing Results.**

| Method | PSNR↑ | SSIM↑ |
|---|---|---|
| TransWeather [65] | 23.11 | 0.8543 |
| SnowFormer [12] | 24.01 | 0.8939 |
| S2VD [78] | 22.95 | 0.8590 |
| RDDNet [68] | 22.97 | 0.8742 |
| EDVR [69] | 17.93 | 0.5790 |
| BasicVSR [6] | 22.46 | 0.8473 |
| IconVSR [6] | 22.35 | 0.8482 |
| BasicVSR++ [7] | 22.64 | 0.8618 |
| RVRT [33] | 20.90 | 0.7974 |
| SVDNet [10] | 25.06 | 0.9210 |
| TURTLE | **26.02** | **0.9230** |

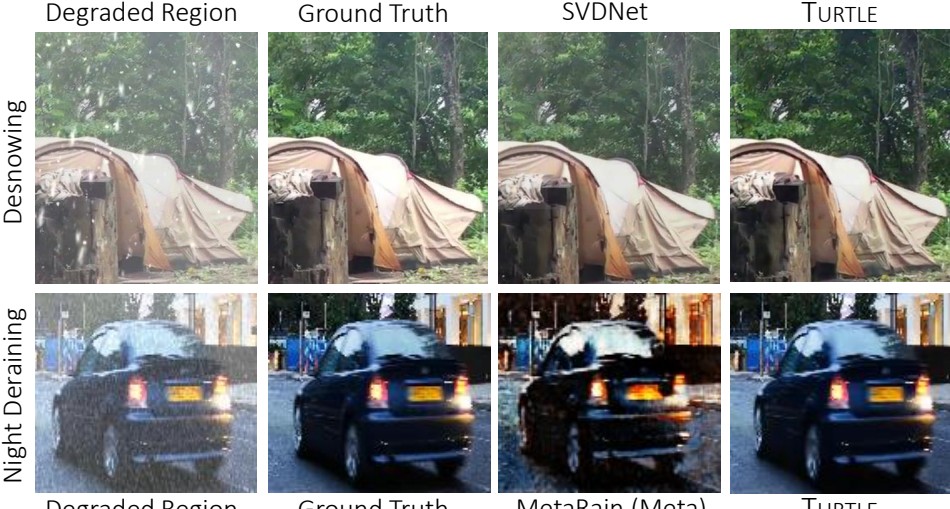

Figure 3: **Visual Results on Video Desnowing and Nighttime Video Deraining.** We compare video desnowing results with the best published method in literature, SVDNet [10]. The video frame has both snow, and haze. While SVDNet [10] removes snow flakes, TURTLE can remove haze, and snow flakes, and hence is more faithful to the ground truth. In nighttime deraining, we compare TURTLE to MetaRain [47]. TURTLE maintains color consistency in the restored result.

compute the output, $\mathbf{y}_t^{[l]}$, as,

$$\mathbf{y}_t^{[l]} = \left[ \sigma(\mathbf{A}_t^{[l]})\mathbf{V}_t^{[l]} \right] W^{\hat{\mathbf{H}}_t^{[l]}} + \mathcal{D}_t(\mathbf{F}_t^{[l]}) \in \mathbb{R}^{h^l \times w^l \times c^l}. \tag{7}$$

In eq. (7), $\psi_t(\hat{\mathbf{H}}_t^{[l]}, \mathbf{F}_t^{[l]}) = \left[ \sigma(\mathbf{A}_t^{[l]})\mathbf{V}_t^{[l]} \right] W^{\hat{\mathbf{H}}_t^{[l]}}$ which follows from eq. (2).

## 4 Experiments

We follow the standard training setting of architectures in the restoration literature [29, 79, 15] with Adam optimizer [27] ($\beta_1 = 0.9, \beta_2 = 0.999$). The initial learning rate is set to $4e^{-4}$, and is decayed to $1e^{-7}$ throughout training following the cosine annealing strategy [40]. All of our models are implemented in the PyTorch library, and are trained on 8 NVIDIA Tesla v100 PCIe 32 GB GPUs for 250k iterations. Each training video is sampled into clips of $\gamma = 5$ frames, and TURTLE restores frames of each clip with recurrence. The training videos are cropped to $192 \times 192$ sized patches at random locations, maintaining temporal consistency, while the evaluation is done on the full frames during inference. We assume no prior knowledge of the degradation process for all the tasks. Further, we apply basic data augmentation techniques, including horizontal-vertical flips and 90-degree rotations. Following the video restoration literature, we use Peak Signal-to-Noise

Table 3: **Real-World Video Deblurring.** Quantitative results (PSNR, and SSIM) on the 3ms-24ms BSD dataset [83] comparing state-of-the-art methods.

| Method | PSNR↑ | SSIM↑ |
|---|---|---|
| STRCNN [24] | 29.42 | 0.893 |
| DBN [58] | 31.21 | 0.922 |
| SRN [60] | 28.92 | 0.882 |
| IFI-RNN [42] | 30.89 | 0.917 |
| STFAN [86] | 29.47 | 0.872 |
| CDVD-TSP [44] | 31.58 | 0.926 |
| PVDNet [57] | 31.35 | 0.923 |
| ESTRNN [83] | 31.39 | 0.926 |
| **TURTLE** | **33.58** | **0.954** |

Table 4: **Synthetic Video Deblurring Results.** Quantitative results (PSNR, and SSIM) on the GoPro dataset [41] comparing state-of-the-art methods.

| Method | PSNR↑ | SSIM↑ |
|---|---|---|
| IFI-RNN [42] | 31.05 | 0.9110 |
| ESTRNN [82] | 31.07 | 0.9023 |
| EDVR [69] | 31.54 | 0.9260 |
| TSP [44] | 31.67 | 0.9280 |
| GSTA [59] | 32.10 | 0.9600 |
| FGST [36] | 32.90 | 0.9610 |
| BasicVSR++ [7] | 34.01 | 0.9520 |
| DSTNet [45] | 34.16 | 0.9679 |
| **TURTLE** | **34.50** | **0.9720** |

Table 5: **Video Raindrop and Rain Streak Removal.** Quantitative results (PSNR, and SSIM) on the VRDS dataset [71] comparing state-of-the-art methods.

| Method | PSNR↑ | SSIM↑ |
|---|---|---|
| S2VD [78] | 18.95 | 0.6630 |
| EDVR [69] | 19.19 | 0.6363 |
| BasicVSR [6] | 28.35 | 0.8990 |
| VRT [34] | 27.77 | 0.8856 |
| TTVSR [37] | 28.05 | 0.8998 |
| RVRT [33] | 28.24 | 0.8857 |
| RDDNet [68] | 28.38 | 0.9096 |
| BasicVSR++ [7] | 29.75 | 0.9171 |
| ViMPNet [71] | 31.02 | 0.9283 |
| **TURTLE** | **32.01** | **0.9590** |

Ratio (PSNR) and Structural Similarity Index (SSIM) [70] distortion metrics to report quantitative performance. For qualitative evaluation, we present visual outputs for each task and compare them with the results obtained from previous best methods in the literature.

## 4.1 Night Video Deraining

SynNightRain [47] is a synthetic video deraining dataset focusing on nighttime videos wherein rain streaks get mixed in with significant noise in low-light regions. Therefore, nighttime deraining with heavy rain is generally a harder restoration task than other daytime video deraining. We follow the train/test protocol outlined in [47, 35], and train TURTLE on 10 videos from scratch, and evaluate on a held-out test set of 20 videos. We report distortion metrics, PSNR and SSIM, in table 1, and compare them with previous restoration methods. TURTLE achieves a PSNR of 29.26 dB, which is a notable improvement of **+2.53** dB over the next best result, NightRain [35]. Further, we present visual results in fig. 3, and in fig. 12. Our method, TURTLE, maintains color consistency in the restored results.

## 4.2 Video Desnowing

Realistic Video Desnowing Dataset (RVSD) [10] is a video-first desnowing dataset simulating realistic physical characteristics of snow and haze. The dataset comprises a variety of scenes, and the videos are captured from various angles to capture realistic scenes with different intensities. In total, the dataset includes 110 videos, of which 80 are used for training, while 30 are held-out test set to measure desnowing performance. We follow the proposed train/test split in the original work [10] and train TURTLE on the video desnowing dataset. Our scores, 26.02 dB in PSNR, are reported in table 2, and compared to previous methods, TURTLE significantly improves the performance by **+0.96** dB in PSNR. Notably, TURTLE is prior-free, unlike the previous best result SVDNet [10], which exploits snow-type priors. We present visual results in fig. 3, and in fig. 11 comparing TURTLE to SVDNet [10]. Our method not only removes snowflakes but also removes haze, and the restored frame is visually pleasing.

## 4.3 Real Video Deblurring

The work done in [83, 82] introduced a real-world deblurring dataset (BSD) using the Beam-Splitter apparatus. The dataset introduced contains three different variants depending on the blur intensity settings. Each of the three variants has a total of $11,000$ blurry/sharp pairs with a resolution of $640 \times 480$. We employ the variant of BSD with the most blur exposure time, i.e., 3ms-24ms.[5] We follow the standard train/test split introduced in [83] with 60 training videos, and 20 test videos. We report the scores in table 3 on the 3ms-24ms variant of BSD and compare with previously published methods. TURTLE scores 33.58 dB in PSNR on the task, observing an increase of **+2.0** dB compared to the previous best methods, CDVD-TSP [44], and ESTRNN [83, 82]. We present visual results in fig. 13.

---

[5]24ms is the blur exposure time, and 3ms is the sharp exposure time.

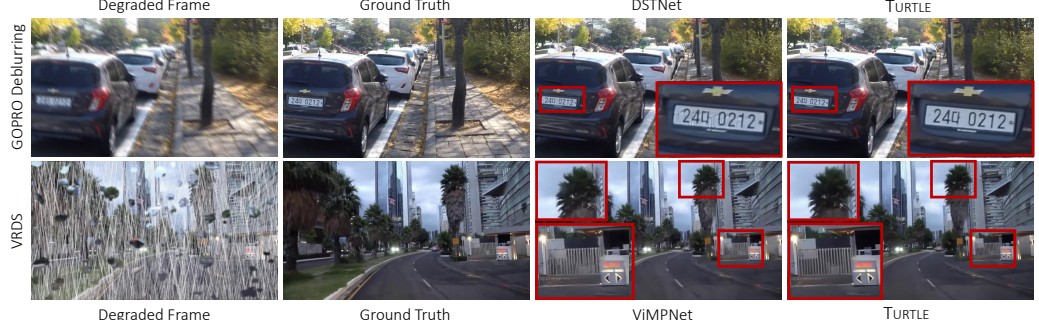

Figure 4: **Visual Results on Video Deblurring and Raindrops and Rain Streaks Removal.**
Qualitative results on video deblurring on the GoPro dataset [41] are in the top row. Our method,
TURTLE, restores the frames without any artifacts (see the number plate) unlike DSTNet [45]. On
video raindrops and rain streaks removal task, we compare our method with the best method in
literature ViMPNet [71]. Notice how the frame restored by ViMPNet [71] has artifacts (see tree
region, and the railing gate), while TURTLE's output is free of unwanted artifacts.

Table 6: **Blind Video Denoising Results.** Quantitative results on blind video denoising task in terms of distortion metrics, PSNR and SSIM, on two datasets DAVIS [48], and Set8 [61].

| Methods | DAVIS | | Set8 | |
|---|---|---|---|---|
| | $\sigma = 30$ | $\sigma = 50$ | $\sigma = 30$ | $\sigma = 50$ |
| VLNB [1] | 33.73 | 31.13 | 31.74 | 29.24 |
| FastDVDNet [62] | 34.04 | 31.86 | 31.60 | 29.42 |
| DVDNet [61] | 34.08 | 31.85 | 31.79 | 29.56 |
| UDVD [55] | 33.92 | 31.70 | 32.01 | 29.89 |
| ReMoNet [72] | 33.93 | 31.65 | 31.59 | 29.44 |
| BSVD-32 [49] | 34.46 | 32.25 | 31.71 | 29.62 |
| BSVD-64 [49] | **34.91** | **32.72** | 32.02 | 29.95 |
| TURTLE | 34.48 | 32.38 | **32.22** | **30.29** |

Table 7: 4× **Video Super Resolution.** Quantitative results on video super resolution task in terms of distortion metrics, PSNR and SSIM.

| Method | PSNR↑ | SSIM↑ |
|---|---|---|
| TDAN [63] | 23.07 | 0.7492 |
| EDVR [69] | 23.51 | 0.7611 |
| BasicVSR [6] | 23.38 | 0.7594 |
| MANA [76] | 23.15 | 0.7513 |
| TTVSR [37] | 23.60 | 0.7686 |
| BasicVSR++ [7] | 23.70 | 0.7713 |
| EAVSR [67] | 23.61 | 0.7618 |
| EAVSR+ [67] | 23.94 | 0.7726 |
| **TURTLE** | **25.30** | **0.8272** |

## 4.4 Synthetic Video Deblurring

GoPro dataset [41] is an established video deblurring benchmark dataset in the literature. The dataset is prepared with videos taken from a GOPRO4 Hero consumer camera, and the videos are captured at 240fps. Blurs of varying strength are introduced in the dataset by averaging several successive frames; hence, the dataset is a synthetic blur dataset. We follow the standard train/test split of the dataset [41], and train our proposed method. TURTLE scores 34.50 dB in PSNR on the task, with an increase of **+0.34** dB compared to the previous best method in a comparable computational budget, DSTNet [45] (see table 4). We also present visual results on the GoPro dataset [41] comparing TURTLE to DSTNet [45] in fig. 4, and fig. 9. Our method restores frames free of artifacts (see the number plate on the car) in fig. 4.

## 4.5 Video Raindrops and Rain Streak Removal

The work done in [71] introduced a synthesized video dataset of 102 videos, VRDS, wherein the videos contain both raindrops and rain streaks degradations since both rain streaks and raindrops corrupt the videos captured in rainy weather.[6] We split the dataset in train and held-out test sets as outlined in the original work [71]. We present TURTLE's scores in table 5, and compare it with several methods in the literature. TURTLE scores 32.01 dB in PSNR on the task, with an increase of **+0.99** dB compared to the previous best method, ViMPNet [71]. We present visual results on the task in fig. 4, and fig. 10, and compare our method with ViMPNet [71]. TURTLE restores the frames that are pleasing to the human eye and are faithful to the ground truth.

---

[6]The dataset comprises videos captured in diverse scenarios in both daytime and nighttime settings.

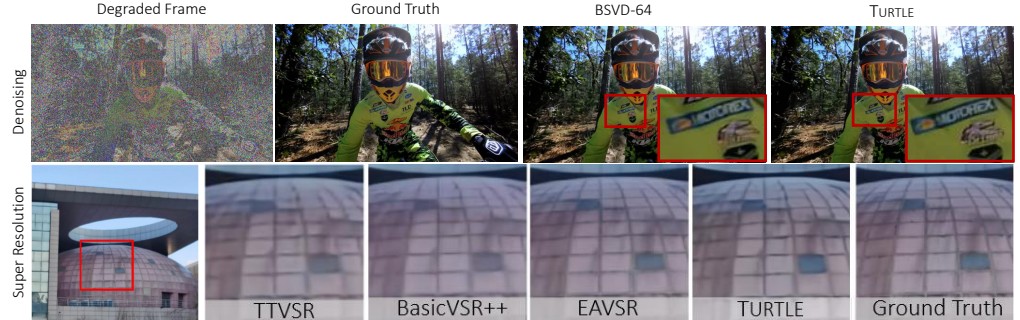

Figure 5: **Blind Video Denoising and Video Super-Resolution Visual Results.** Qualitative comparison of previous methods with TURTLE on a test frame from Set8 dataset for blind video denoising ($\sigma = 50$), and MVSR4× dataset [71] for video super resolution. In video denoising, TURTLE restores details, while BSVD-64 [49] smudges textures (text and the dinosaur on the biker's jacket). In VSR, previous methods such as TTVSR [37], BasicVSR++ [7], or EAVSR [71] tend to introduce blur in results, while TURTLE's restored results are sharper, and crisper.

Table 8: **MACs (G) Comparison.** We report MACs (G) of TURTLE, and compare with previous methods in literature. We also extensively profile TURTLE with varying input resolutions on a single GPU, and compare it with previous restoration methods in appendix F.

| Method | Venue | Task | MACs (G) ↓ |
|---|---|---|---|
| RVRT [33] | NeurIPS'22 | Restoration | 1182.16 |
| VRT [34] | TIP'24 | Restoration | 1631.67 |
| RDDNet [68] | ECCV'22 | Deraining | 362.36 |
| DSTNet [45] | CVPR'23 | Deblurring | 720.28 |
| EDVR [69] | CVPR'19 | Deraining | 527.5 |
| BasicVSR [6, 7] | CVPR'21 | Super Resolution | 240.17 |
| **TURTLE** | NeurIPS'24 | Restoration | **181.06** |

## 4.6 Video Super-Resolution

MVSR4× is a real-world paired video super-resolution dataset [67] collected by mobile phone's dual cameras. We train TURTLE following the dataset split in the official work [67] and test on the provided held-out test set. We report distortion metrics in table 7 and compare it with several methods in the literature. TURTLE scores 25.30 dB in PSNR on the task, with a significant increase of **+1.36** dB compared to the previous best method, EAVSR+ [71]. We present visual results on the task in fig. 5. Other methods such as TTVSR [37], BasicVSR [7], or EAVSR [71] tend to introduce blur in up-scaled results, while TURTLE's restored results are sharper.

## 4.7 Blind Video Denoising

We assume no degradation prior, and consider blind video denoising task [49, 55]. We train our model on DAVIS [48] dataset, and test on DAVIS held-out testset, and a generalization set Set8 [61]. We add white Gaussian noise to the dataset with noise level $\sigma \in \mathcal{U}[30, 50]$ to train TURTLE, and test on two noise levels $\sigma = 30$, and $\sigma = 50$; scores are reported in table 6. TURTLE observes a gain of **+0.31** dB on $\sigma = 30$, and **+0.34** dB on $\sigma = 50$ on Set8 testset, scoring 32.22 dB, and 30.29 dB, respectively, while it observes an average drop of $-0.3$ dB to BSVD-64 [49] on the DAVIS testset. Further, we present qualitative results in fig. 5 comparing TURTLE, and previous best method BSVD [49].

## 4.8 Computational Cost Comparison

In table 8, we compare TURTLE with previous methods in the literature in terms of multiply-accumulate operations (MACs). The results are computed for the input size $256 \times 256$. We measure the performance on the number of frames the original works utilized[7] to report their performance, as

---

[7] or 2 frames if the actual number of frames did not fit into a single 32GB GPU

Table 9: **State Align Block.**

| Block Configuration | PSNR ↑ |
|---|---|
| No CHM | 31.84 |
| No $\phi$ | 32.07 |
| **TURTLE** | **32.26** |

Table 10: **Truncation Factor.**

| Truncation Factor $\tau$ | PSNR ↑ |
|---|---|
| 1 | 32.15 |
| 5 | 32.26 |
| **TURTLE** ($\tau = 3$) | **32.26** |

Table 11: **Value of** *topk*.

| Value of $k$ in *topk* | PSNR ↑ |
|---|---|
| 1 | 32.18 |
| 20 | 32.10 |
| **TURTLE** ($k = 5$) | **32.26** |

reported in their manuscript or code bases. In TURTLE's case, we report MACs (G) on a single frame since TURTLE only considers a single frame at a time but adjust for history features utilized in CHM as part of TURTLE's decoder. In comparison to parallel methods, EDVR [69], VRT [34], TURTLE is computationally efficient, as it is lower in MACs (G). Although the MACs are approximately similar to recurrent methods, BasicVSR [6], TURTLE scores significantly higher in PSNR/SSIM metrics (see table 7, and table 5). In comparison to contemporary methods such as RVRT [33], which combines recurrence and parallelism in design, TURTLE is significantly lower on MACs (G) and performs better (see table 5, and table 2) thanks to its ability to memorize previous frames.

## 5  Ablation Study

We ablate TURTLE to understand what components necessitate efficiency and performance gains. All experiments are conducted on synthetic video deblurring task, GoPro dataset [41], using a smaller variant of our model. Our smaller models operate within a computational budget of approximately 5 MACs (G), while the remaining settings are the same as those of the main model. In all the cases, the combinations we adopt for TURTLE are highlighted . Additional ablation studies are deferred to appendix A, and we discuss the limitations of the proposed method in appendix C.

**Block Configuration.**    We ablate the Causal History Model (CHM) to understand if learning from history benefits the restoration performance. We compare TURTLE with two settings: baseline (no CHM block) and TURTLE without State Align Block ($\phi$). In baseline (no CHM), no history states are considered, and two frames are concatenated and fed to the network directly. Further, in No $\phi$, the state align block is removed from CHM. We detail the results in table 9, and find that both State Align Block, and CHM are important to the observed performance gains.

**Truncation Factor $\tau$.**    We evaluate context lengths of $\tau = 1$, 3, and 5 past frames and found no PSNR improvement when increasing the context length beyond three frames. Results in table 10 confirm that extending beyond three frames does not benefit performance. This is because, as in most cases, the missing information in the current frame is typically covered within the three-frame span, and additional explicit frame information fails to provide additional relevant details.

**Value of $k$ in *topk*.**    We investigate the effects of different $k$ values in *topk* attention. Our experiments, detailed in table 11, show that $k$ crucially affects restoration quality. Utilizing a larger number of patches, $k = 20$, leads to an accumulation of irrelevant information, negatively impacting performance by adding unnecessary noise. Further, selecting only 1 patch is also sub-optimal as the degraded nature of inputs can lead to inaccuracies in identifying the most similar patch, missing vital contextual information. The optimal balance was found empirically with $k = 5$, which effectively minimizes noise while ensuring the inclusion of key information.

## 6  Conclusion

In this work, we introduced a novel framework, TURTLE, for video restoration. TURTLE learns to restore any given frame by conditioning the restoration procedure on the frame history. Further, it compensates the history for motion with respect to the input and accentuates key information to benefit from temporal redundancies in the sequence. TURTLE enjoys training parallelism and maintains the entire frame history implicitly during inference. We evaluated the effectiveness of the proposed method and reported state-of-the-art results on seven video restoration tasks.

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

**Technical Appendices**

**Contents**

In appendices, we discuss additional details about the proposed method TURTLE, and provide additional ablation studies in appendix A, motivate the need for learning to model the history of input for video restoration in appendix B, discuss limitations of the proposed approach appendix C, discuss theoretical relationship to state-space models in appendix D, present more visual results in appendix E, discuss related work in appendix G, and computationally profile the proposed method TURTLE in appendix F.

<table>
<tr><td colspan="2">Table 12: CHM Placement.</td></tr>
<tr><td>CHM
Placement</td><td>PSNR ↑</td></tr>
<tr><td>in Latent</td><td>32.05</td></tr>
<tr><td>in Latent & Decoder</td><td>32.26</td></tr>
</table>

<table>
<tr><td colspan="2">Table 13: Softmax Ablation.</td></tr>
<tr><td>Ablating
Softmax</td><td>PSNR ↑</td></tr>
<tr><td>Softmax</td><td>32.04</td></tr>
<tr><td>topk (k = 5)</td><td>32.26</td></tr>
</table>

## A   Additional Ablation Studies

We ablate two more aspects of the proposed method TURTLE. Mainly, we empirically verify the rationale behind placing CHM in both latent, and decoder stages. Further, we ablate if selecting *topk* regions compared to plain softmax is beneficial for restoration.

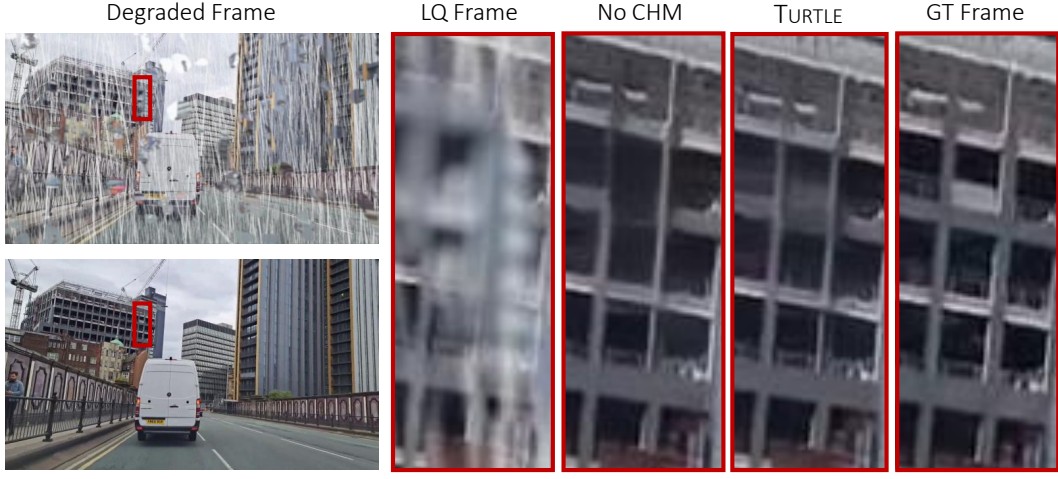

Figure 6: **Do we need history?** We present visual results of TURTLE and TURTLE without CHM to motivate the need for summarizing the history and conditioning the restoration on the history of the input. Other than efficiency, it also brings perceptual benefits. Notice how "no CHM" introduces smudges and blemishes in place of the guard railing in the balcony of the building since the region is obscured in the degraded input.

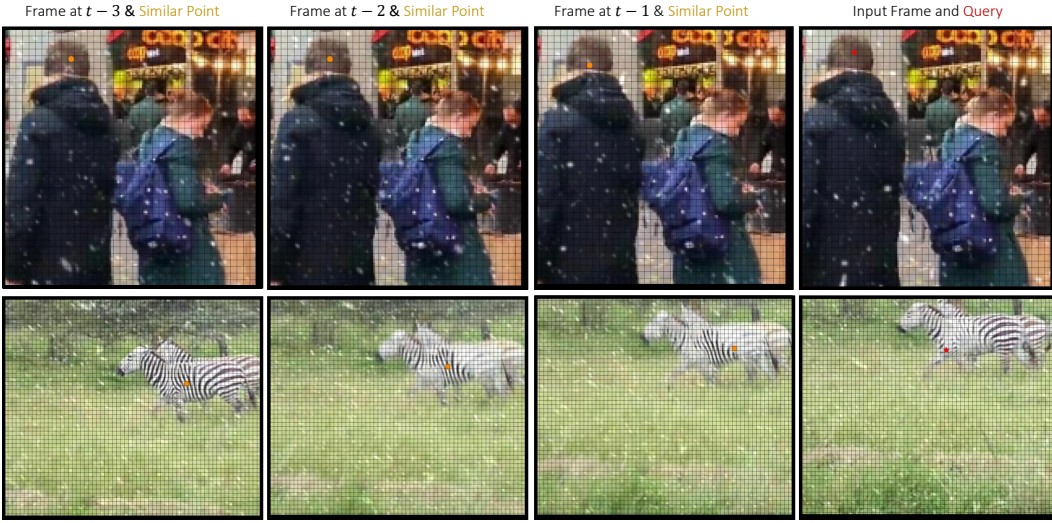

Figure 7: **CHM Tracking.** Visual illustration of CHM tracking query points in the frame history (frames previous to the input frame). In the top row, we plot the correctly tracked points, while the bottom row visualizes the limitations in the case of redundant patterns. We plot the query and most similar points on input frames for ease of exposition, but in practice, they function on feature maps.

**CHM Placement.** Our experiment, in table 12, indicates that having CHM in both the latent and decoder stages is necessary for optimal performance. In the latent stage, the spatial resolution is minimal, and CHM provides greater benefit in the following decoder stages as the spatial resolution increases.

**Softmax Ablation.** In table 13, we verify that *topk* selection is necessary to allow the restoration procedure to only consider relevant information from history. Since softmax does not bound the information flow we observe non-trivial performance drop when *topk* is replaced with softmax. We

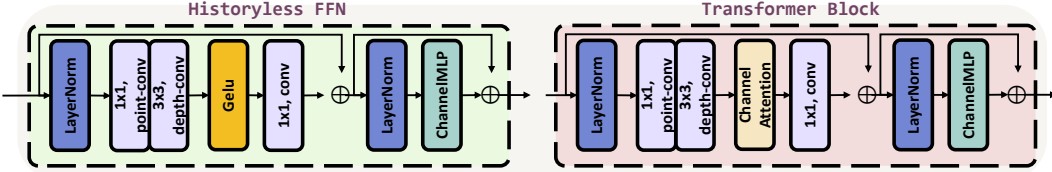

Figure 8: **Illustration of Historyless FFN.** Transformer block is similar in spirit to the block introduced in [79], while the Historyless FFN's design takes inspiration from the blocks in [15, 11].

argue that *topk* prevents the inclusion of unrelated patches, which can, potentially, introduce irrelevant correlations, and obscure principal features.

## B TURTLE's Specifications & Details

We motivate TURTLE's design and present empirical results to pinpoint the benefits of modeling the history in the case of video restoration. Further, we present additional details of the proposed method, TURTLE, and expand on the construction of the architecture.

### B.1 Motivation: Causal History Model

Recall that the Causal History Model (CHM) is designed to model the state and compensate for motion across the entire history relative to the input. It then learns to control the effect of history frames by scoring and aggregating motion-compensated features based on their relevance to the restoration of the current frame. Such a procedure allows for borrowing information from the preceding temporal neighbors of the input frame. In table 9, we ablate if CHM indeed provides the performance boost. Moreover, in fig. 6, we present visual results on the video raindrops and rain streaks removal task to motivate the need for summarizing the frame history as part of the restoration method. We train TURTLE without the CHM block, referred to as "no CHM", following TURTLE's experimental setup, and for fair comparison, we keep the model size consistent. TURTLE, equipped with CHM, maintains the spatial integrity of the input without introducing faux textures or blur even though the region (see guard railing in the balcony of the building) is entirely obscured by raindrops and streaks in the degraded input. However, without CHM block, unwanted artifacts (such as holes and blemishes in place of the guard railing in the balcony) are introduced to fill in the missing information since no information is borrowed from preceding frames. Note that in the case of no CHM experiment, we feed two concatenated frames (one frame previous to the input and the input frame) to the architecture.

### B.2 Historyless FFN.

Recall that TURTLE's encoder is historyless, i.e., it employs no information about the history of the input. Further, we opt for a feedforward style design in the encoder with convolutional layers. This is because shallow representations at this stage are not sufficiently compressed and are riddled with degradation. Thus, expensive attention-based operators provide no significant performance benefit but add to the computational complexity. The diagrammatic illustration of Historyless FFN and the Transformer block [79] used in CHM is presented in fig. 8.

## C Limitations & Discussion

This section discusses the limitations of our proposed CHM. While CHM is adept at tracking similar patches across the history of frames, it encounters challenges in certain scenarios. For instance, as demonstrated in the zebra example in fig. 7, CHM identifies similar patches on different parts of the zebra's body due to their redundant patterns, even though these patches are not located in the same area. Moreover, in case of severe input degradation, CHM's capacity to accurately identify and utilize similar patches may diminish due to spurious correlations, which could affect its ability to use history effectively for restoring the current frame.

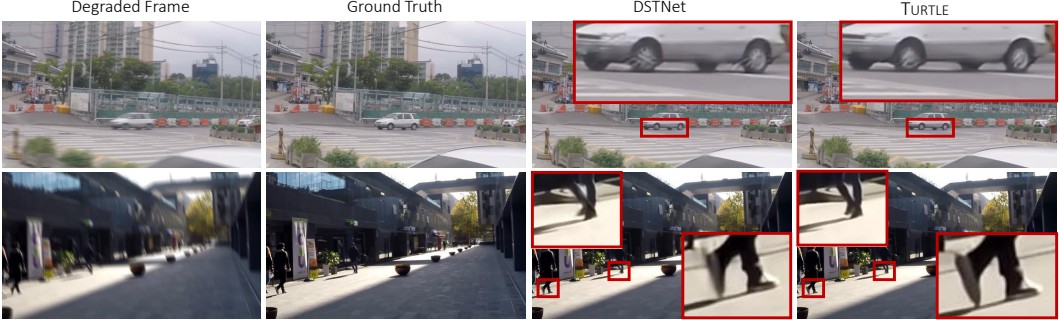

| Degraded Frame | Ground Truth | DSTNet | TURTLE |
|---|---|---|---|

Figure 9: **Visual Results on Synthetic Video Deblurring.** We present additional qualitative analysis on synthetic video deblurring on the GoPro dataset [41]. We compare TURTLE with DSTNet [45] on two frames taken from two different videos of the testset.

### C.1 Societal Impact

This work presents an efficient method to advance the study of machine learning research for video restoration. While the proposed method effectively restores the degraded videos, we recommend expert supervision in medical, forensic, or other similar sensitive applications.

## D Relationship to State Space Models

We present a special case of the proposed Causal History Model, CHM, wherein the videos are not degraded, and each frame is optimally compensated for motion with respect to the input.

**Lemma D.1.** *(Special Case of Causal History Model) In the absence of degradation and optimally compensated motion through optical flow, the state history $\hat{\mathbf{H}}_t^{[l]}$, then, only depends on the input $\mathbf{F}_t^{[l]}$, and the previous state $\hat{\mathbf{H}}_{t-1}^{[l]}$. Under this assumption, eq. (1), and eq. (2) can be rewritten as,*

$$\hat{\mathbf{H}}_t^{[l]} = \mathbf{A}_t(\hat{\mathbf{H}}_{t-1}^{[l]}) + \mathbf{B}_t(\mathbf{F}_t^{[l]}), \tag{8}$$

$$\mathbf{y}_t^{[l]} = \mathbf{C}_t(\hat{\mathbf{H}}_t^{[l]}), \tag{9}$$

where $\mathbf{A}_t$, $\mathbf{B}_t$, $\mathbf{C}_t$ are parameters learned through gradient descent, and $\mathbf{F}_t^{[l]}$ is the input feature map at timestep $t$. In this case, eq. (8) is realizable and not flawed, given the motion-compensated input. This assumption allows the model to be learned in a similar fashion to HiPPO [18] or Mamba [17]. More specifically, the Causal History Model (CHM) reduces to an equivalent time-variant and input-dependent flavor of the State Space Model (SSM).

*Proof 1.* Consider a state space model (SSM) [16] that maps the input signal $\mathbf{F}_t$ to the output signal $\mathbf{y}_t$ through an implicit state $\mathbf{H}_t$, i.e.,

$$\mathbf{H}_t = \mathbf{A}(\mathbf{H}_{t-1}) + \mathbf{B}(\mathbf{F}_t), \tag{10}$$

$$\mathbf{y}_t = \mathbf{C}(\mathbf{H}_t). \tag{11}$$

In the above equations, we abuse the SSM notation for exposition. Recall that we consider the special case wherein the motion is compensated for, and there is no degradation in the input video. In this case, we can say that history and motion-compensated history are equal i.e., $\hat{\mathbf{H}}_t = \mathbf{H}_t$. Now, consider the first frame of the video at timestep $t = 0$. Let the initial condition be denoted by $\mathbf{F}_0$, then we can write the eq. (10), and eq. (11) as,

$$\hat{\mathbf{H}}_0 = \mathbf{B}_0(\mathbf{F}_0), \quad \text{because} \quad \mathbf{H}_{t-1} = \mathbf{0} \tag{12}$$

$$\mathbf{y}_0 = \mathbf{C}_0(\hat{\mathbf{H}}_0), \tag{13}$$

where $\mathbf{A}_0$, $\mathbf{B}_0$, $\mathbf{C}_0$ are learnable parameters. Then for the next timestep $t = 1$, we can write that

$$\hat{\mathbf{H}}_1 = \mathbf{A}_1(\hat{\mathbf{H}}_0) + \mathbf{B}_1(\mathbf{F}_1), \tag{14}$$

$$\mathbf{y}_1 = \mathbf{C}_1(\hat{\mathbf{H}}_1). \tag{15}$$

From eq. (12), we know that $\hat{\mathbf{H}}_0 = \mathbf{B}_0(\mathbf{F}_0)$, then we can re-write eq. (14) as

$$\hat{\mathbf{H}}_1 = \mathbf{A}_1(\mathbf{B}_0(\mathbf{F}_0)) + \mathbf{B}_1(\mathbf{F}_1) \tag{16}$$
$$\Rightarrow \mathbf{A}_1\mathbf{B}_0(\mathbf{F}_0) + \mathbf{B}_1(\mathbf{F}_1).$$

The output $\mathbf{y}_1$ can then be written as,

$$\mathbf{y}_1 = \mathbf{C}_1(\hat{\mathbf{H}}_1). \tag{17}$$

At every timestep $t$, the output can be written in terms of the history (previous timestep) and the input (at current timestep). Now, consider the case $t = 2$, and let the frame features be denoted by $\mathbf{F}_2$, then we can write,

$$\hat{\mathbf{H}}_2 = \mathbf{A}_2(\hat{\mathbf{H}}_1) + \mathbf{B}_2(\mathbf{F}_2), \tag{18}$$
$$\Rightarrow \mathbf{A}_2(\mathbf{A}_1(\mathbf{B}_0(\mathbf{F}_0))) + \mathbf{B}_1(\mathbf{F}_1) + \mathbf{B}_2(\mathbf{F}_2), \quad \text{because } eq.\ (16)$$

$$\text{therefore,} \quad \hat{\mathbf{H}}_2 = \underbrace{\mathbf{A}_2\mathbf{A}_1\mathbf{B}_0(\mathbf{F}_0) + \mathbf{B}_1(\mathbf{F}_1)}_{\text{History}} + \underbrace{\mathbf{B}_2(\mathbf{F}_2)}_{\text{Input}}. \tag{19}$$

Notice, how in eq. (19) $\hat{\mathbf{H}}_2$ is written in terms of the input frame $\mathbf{F}_2$, and the previous frames $\mathbf{F}_0$, and $\mathbf{F}_1$. The output $\mathbf{y}_2$ is then computed as,

$$\mathbf{y}_2 = \mathbf{C}_2(\hat{\mathbf{H}}_2). \tag{20}$$

We can then generalize eq. (18), and eq. (20) to any timestep $t$, and we arrive at eq. (8), and eq. (9), i.e.,

$$\hat{\mathbf{H}}_t = \mathbf{A}_t(\hat{\mathbf{H}}_{t-1}) + \mathbf{B}_t(\mathbf{F}_t), \tag{21}$$
$$\mathbf{y}_t = \mathbf{C}_t(\hat{\mathbf{H}}_t). \tag{22}$$

Therefore, the model can be learned in Mamba [17], or HiPPO [18] style, and the parameters $\mathbf{A}$, $\mathbf{B}$, $\mathbf{C}$ can be learned with gradient descent since the linear relationship between the input and the output is tractable in this case. □

In practice, however, the no degradation assumption does not hold. Therefore, a naive state update, eq. (8), renders sub-optimal results. This is because, in video processing tasks, motion governs the transition dynamics, i.e., the state evolves over time due to motion, and therefore, any linear relationship between the output and the input is intractable unless the motion is compensated for.

## E    Further Visual Comparisons

We present additional visual results, comparing our method with previously available methods in the video restoration literature.

### E.1    Synthetic Video Deblurring

We present further results on synthetic video deblurring task on the GoPro dataset [41] in fig. 9. We compare TURTLE with the previous method in the literature that is computationally similar to TURTLE, DSTNet [45]. Turtle avoids unnecessary artifacts in the restored results (see the tire undergoing rotation in the frame in the top row in fig. 9). Further, the restored results are not smudged, and textures are restored faithfully to the ground truth (see feet of the person in the frame in the bottom row).

### E.2    Video Raindrops and Rain Streaks Removal

Unlike just rain streaks, raindrops often pose a more complex challenge for restoration algorithms. This is because several video/image restoration methods often induce blurriness in the results. Raindrops get mixed in with the background textures; therefore, minute details such as numbers or text are blurred in restored results. However, since TURTLE utilizes the neighboring information, it learns to restore these details better. We observe this in fig. 10, where the previous best method ViMPNet [71] blurs the number plate on the car (in last row), or introduces faux texture on the building (in second row). On the contrary, TURTLE better restores the results and avoids undesired artifacts or blur.

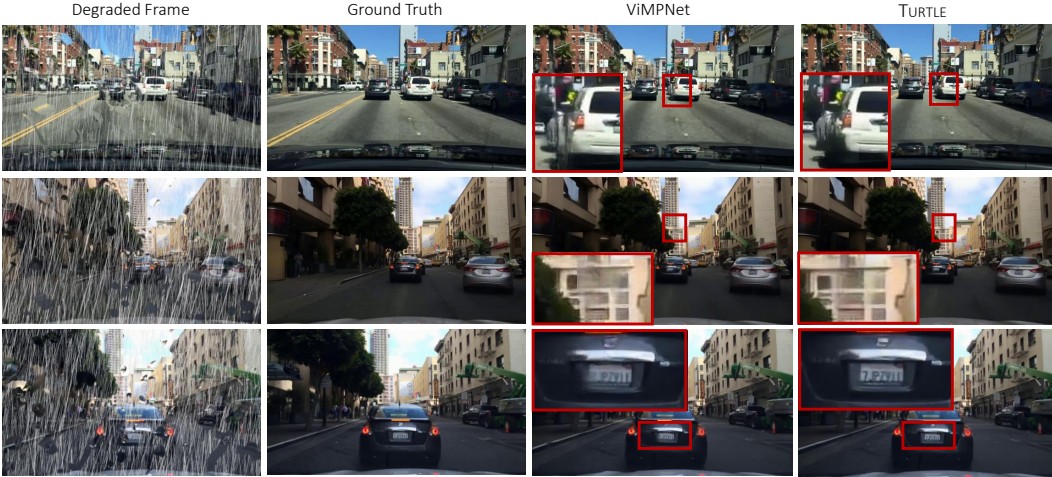

Figure 10: **Additional Visual Results on VRDS.** We present additional qualitative analysis on video raindrops and rain streaks removal (VRDS) tasks. We compare TURTLE with ViMPNet [71], the best method in literature, on three frames taken from the testset. TURTLE's results are artifacts-free as it can effectively remove both the streaks and drops. However, ViMPNet [71] tends to mix in the raindrops with the background, introducing smudge (see the building) and blur (see number plate on the car).

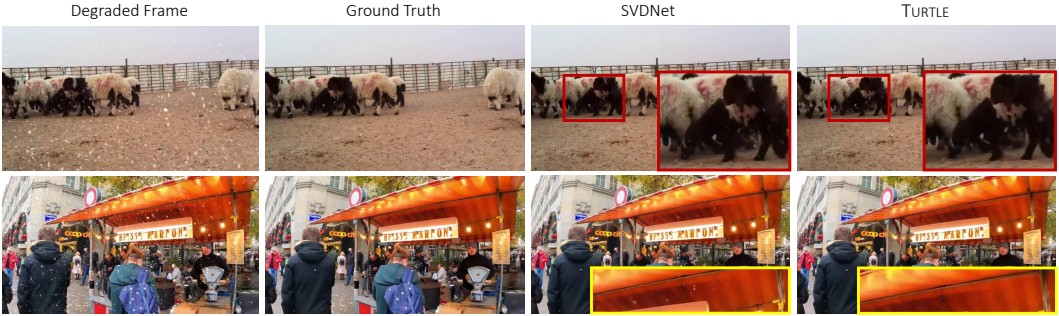

Figure 11: **Additional Visual Results on RSVD.** We present additional qualitative analysis on the desnowing task. We compare TURTLE with SVDNet [10], the best method in literature, on two frames taken from the testset. TURTLE removes even smaller snowflakes (flecks of snow on the underside of orange roof) and differentiates between textures and snow (see white spots on sheep).

### E.3 Video Desnowing

We present additional visual results on the video desnowing task in fig. 11. We compare TURTLE with SVDNet [10], the previous best method in the literature on the task. Our method removes snowflakes effectively and differentiates between them and similar-sized texture regions in the background (see white spots on the sheep wool in the top row) without requiring any snow prior like SVDNet [10]. Although SVDNet [10] removes snowflakes to a considerable extent, it fails to remove smaller flecks of snow comprehensively. However, TURTLE's restored results are visually pleasing and faithful to the ground truth.

### E.4 Nighttime Video Deraining

In fig. 3, we presented the visual results in comparison to MetaRain [47]. For a fair comparison, we resized the outputs to $256 \times 256$ following the work in [47, 35]. However, in fig. 12, we present TURTLE's results on full-sized frames taken from two different testset videos. Our method preserves the true colors of the frames and removes rain streaks from the input without introducing

| Degraded Frame | Ground Truth | TURTLE |

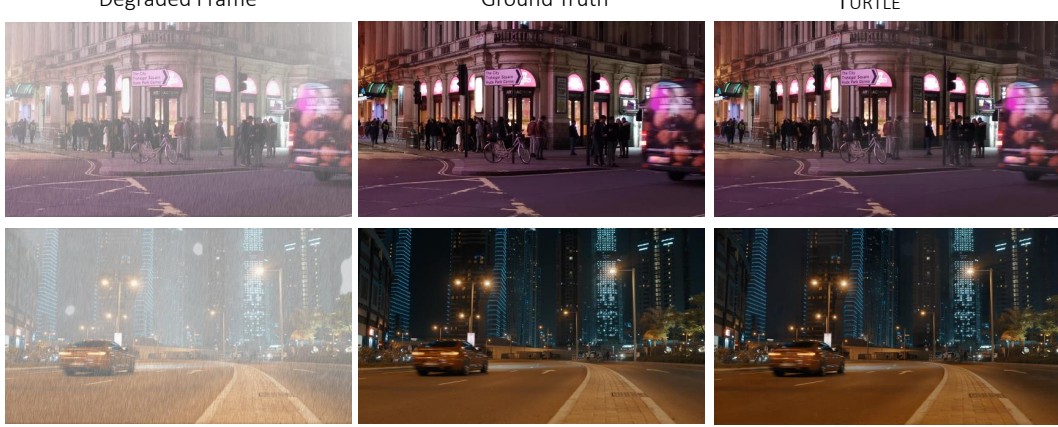

Figure 12: **Additional Visual Results on Nighttime Deraining.** We present additional visual results on the nighttime deraining dataset [47]. TURTLE maintains color consistency, is artifact-free, and is more faithful to the ground truth.

unwanted textures or discoloration. Note that in table 1, we compute PSNR in Y-Channel following MetaRain [47] since NightRain [35] did not release their code, and their manuscript does not clarify if the scores are computed in RGB color space or in Y-Channel. Nonetheless, we report PSNR score in RGB color space to allow comparison with NightRain [35] regardless: TURTLE scores 27.68 dB in RGB color space.

### E.5 Real-World Video Deblurring

Different from synthetic video deblurring (see fig. 9), in real-world deblurring, the blur is induced through real motion, both of camera, and object. In fig. 13, we present visual results on three frames taken from three videos from the testset of the BSD dataset (3ms-24ms configuration) to complement the quantitative performance of TURTLE in table 3. TURTLE restores the video frames with high perceptual quality, and the resultant frames are faithful to the ground-truth, and are visually pleasing.

### E.6 Real-World Weather Degradations

In fig. 14, we present qualitative results of TURTLE on real-world weather degradations. The purpose of these results is to verify generalizability of TURTLE on non-synthetic degradations. Therefore, in addition to real-world video superresolution (see fig. 5), and real-world deblurring (see fig. 13), we also consider real-world weather degradations. We download four videos randomly chosen from a free stock video website. First two videos (in first two columns) are afflicted by snow degradation, while the last two are by rain. Notice how in the last video (in the last column), there is also haze that affects the video. TURTLE removes snow, and rain (including haze) and the restored frames are visually pleasing. Given the lack of ground-truth in these videos, we do not report any quantitative performance metric.

## F   Computational Profile of TURTLE

In table 14, we report the runtime analysis of the proposed method TURTLE on a single 32GB V100 GPU, and compare it with three representative general video restoration methods in the literature, namely ShiftNet [29], VRT [34], and RVRT [33]. We consider four different input resolutions varying from $(256 \times 256 \times 3)$ to 1080p. Prior methods exhibit exponential growth in GPU memory requirements as the resolution increases, TURTLE, however, features linear scaling in GPU memory usage, underscoring its computational efficiency advantage. As the resolution increases beyond 480p, all of the previous methods throw OOM (Out-of-Memory) errors indicating that the memory requirement exceeded the total available memory (of 32GB). On the flip side, TURTLE can process the videos even at a resolution of 1080p on the same GPU.

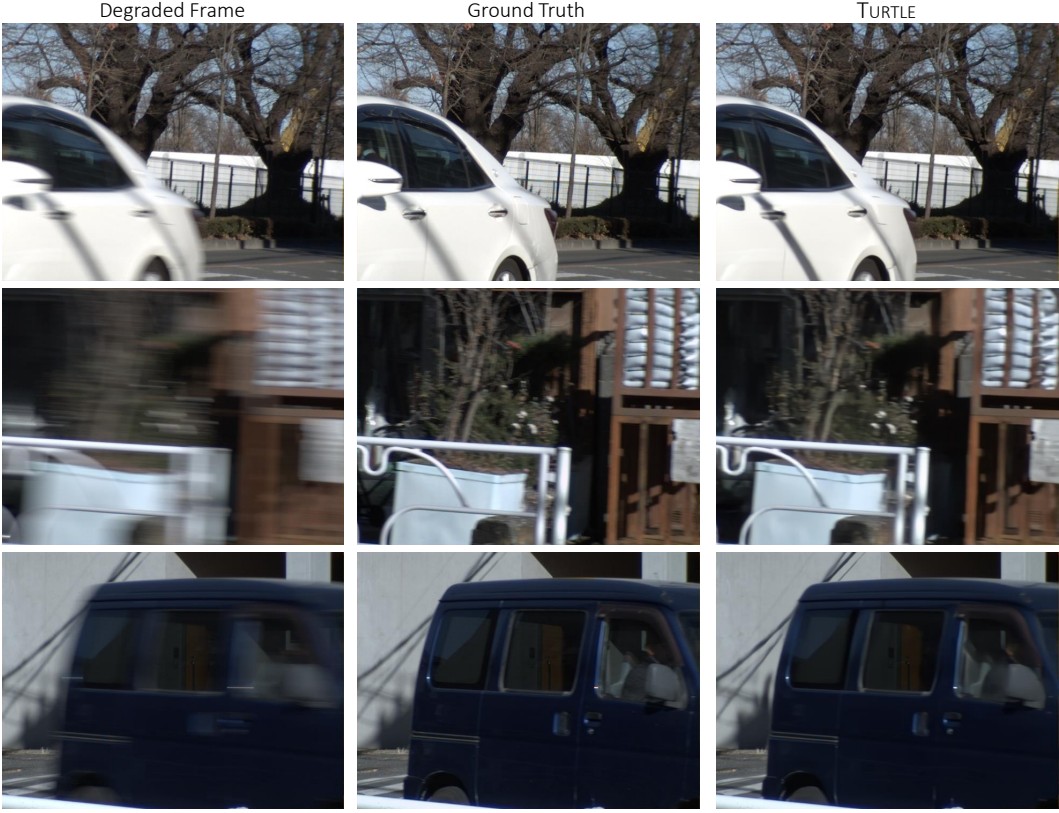

Figure 13: **Visual Results on Real-World Video Deblurring.** We present visual results of TURTLE on the real-world video deblurring task on BSD 3ms-24ms dataset [83, 82] on three frames taken from three different videos in the testset. Our method restores the frame with high perceptual quality.

## G    Additional Literature Review

We further the discussion on prior art in the literature in terms of temporal modeling of videos, and causal learning.

**Temporal Modeling.**    In video restoration, temporal modeling mainly focuses on how the neighboring frames (either in history or in the future) can be utilized to restore the current frame better. For such a procedure, the first step usually involves compensating for motion either through explicit methods (such as using optical flow [33, 34, 36, 7]), or implicitly (such as deformable convolutions [63], search approaches [64], or temporal shift [29]). A few works in the literature focus on reasoning at the trajectory level (i.e., considering the entire frame sequence of a video) [37] through learning to form trajectories of each pixel (or some group of pixels). The motivation is that in this case, each pixel can borrow information from the entire trajectory instead of focusing on a limited context. The second step is then aggregating such information, where in the case of Transformers, attention is employed, while MLPs are also used in other cases.

**Causal Learning in Videos.**    In videos, causal learning is generally explored in the context of self-supervised learning to learn representations from long-context videos with downstream applications to various video tasks such as action recognition, activity understanding, etc [50, 73]. In [2], causal masking of several frames at various spatio-temporal regions as a strategy to learn the representations is explored. To the best of our knowledge, other than one streaming video denoising method [49], almost all of the state-of-the-art video restoration methods are not causal by design since they rely on forward and backward feature propagation (i.e., they consider both frames in history and in the future) either aligned with the optical flow or otherwise [6, 7, 33, 29]. However, there is significant

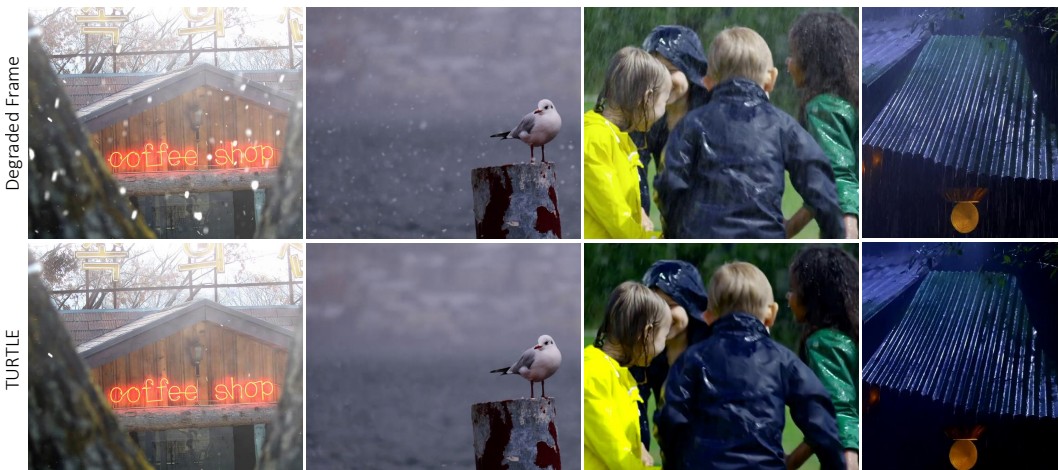

Figure 14: **Real-World Weather Degradations.** We present visual results on real-world weather degradations. The samples are taken from four different videos downloaded from a free stock video website (www.pexels.com). The first two columns contain frames from videos afflicted by snow degradation, while the last two are afflicted by rain degradations. TURTLE restores the frames reliably, and the resultant frames are pleasing to the eye.

Table 14: **Profiling TURTLE.** We profile the proposed method, TURTLE, on a single 32 GB V100 GPU, and compare with 3 recent video restoration methods, namely ShiftNet [29], VRT [34], and RVRT [33]. We consider different input resolutions and compute the per-frame inference time (ms), total MACs (G), FLOPs (G), and the GPU memory usage of the model. OOM denotes Out-Of-Memory error i.e., the memory requirement exceeded the total available memory of 32GB.

| Methods | Frame Resolution | Per Frame Inference Time (ms) | MACs (G) | GPU Memory Usage (MBs) |
|---|---|---|---|---|
| ShiftNet [29] | $256 \times 256 \times 3$ | 190 | 989 | 2752 |
| | $640 \times 480 \times 3$ | 510 | 5630 | 7068 |
| | $1280 \times 720 \times 3$ | OOM | OOM | OOM |
| | $1920 \times 1080 \times 3$ | OOM | OOM | OOM |
| VRT [34] | $256 \times 256 \times 3$ | 455 | 1631 | 3546 |
| | $640 \times 480 \times 3$ | 2090 | 7648 | 11964 |
| | $1280 \times 720 \times 3$ | OOM | OOM | OOM |
| | $1920 \times 1080 \times 3$ | OOM | OOM | OOM |
| RVRT [33] | $256 \times 256 \times 3$ | 252 | 1182 | 5480 |
| | $640 \times 480 \times 3$ | 1240 | 10588 | 21456 |
| | $1280 \times 720 \times 3$ | OOM | OOM | OOM |
| | $1920 \times 1080 \times 3$ | OOM | OOM | OOM |
| **TURTLE** | $256 \times 256 \times 3$ | 95 | 181 | 2004 |
| | $640 \times 480 \times 3$ | 380 | 812 | 4826 |
| | $1280 \times 720 \times 3$ | 1180 | 2490 | 11994 |
| | $1920 \times 1080 \times 3$ | 2690 | 5527 | 24938 |

amount of work on causal representation learning where the aim is to recover the process generating the data from the observation to learn the disentangled latent representation. Note that this is out of the scope of this work.

## H  Dataset Information & Summary

All of the experiments presented in this manuscript employ publicly available datasets that are disseminated for the purpose of scientific research on video/image restoration. All of the datasets

employed are cited wherever they are referred to in the manuscript, and we summarize the details here.

- Video Desnowing: We utilize the video desnowing dataset introduced in [10]. The dataset is made available by the authors at the link: Video Desnowing

- Video Nighttime Deraining: We utilize the nighttime video deraining dataset introduced in [47]. The dataset is made available by the authors at the link: Nighttime Deraining

- Video Raindrops and Rain Streaks Removal: We utilize the video raindrops and rain streaks removal (VRDS) dataset introduced in [71]. The dataset is made available by the authors at the link: VRDS

- Synthetic Video Deblurring: We employ the GOPRO dataset introduced in [41]. The dataset is made available by the authors at the link: GOPRO

- Real Video Deblurring: We employ the BSD dataset introduced in [82, 83]. The dataset is made available by the authors at the link: BSD

- Real-World Video Super Resolution: We utilize the MVSR dataset introduced in [67]. The dataset is made available by the authors at the link: MVSR

- Video Denoising: We employ DAVIS [48], and Set8 [61] datasets for video denoising. The datasets are available at: DAVIS-2017, Set8 [4 sequences], and Set8 [4 sequences]

