# OpenReview forum: "Learning Truncated Causal History Model for Video Restoration"
_NeurIPS.cc/2024/Conference — NeurIPS 2024 poster_

### Official Review · Reviewer_PswG · 2024-06-23

**Soundness:** 4
**Presentation:** 4
**Contribution:** 3
**Rating:** 7
**Confidence:** 4

**Summary:**

This paper proposes a TURTLE to learn the truncated causal history model for video restoration tasks.
The proposed turtle's causal history model consists of two sub-modules: a State Align Block and a Frame History Router.
The state align block has a similarity-based retrieval mechanism that implicitly accounts for inter-frame motion and alignment from previous frames, and those retrieved representations are summarized and stored into a truncated history.
Then the frame history router generates output frames by cross-frame channel attention with the motion-compensated history states.
In experiments, the proposed method shows a promising result in a number of video restoration tasks including deraining, desnowing, deblurring, and super-resolution.

**Strengths:**

- The proposed methods outperforms others in a number of video restoration tasks and benchmarks.

- The proposed causal history model has a capability to consider spatio-temporal-channel correlation by the state align block (spatio-temporal) and the frame history router (channel-temporal).

- It theoretically shows the link with the space state model.

- It seems to increase efficiency by designing the encoder to process single frame only while the decoder to process multiple frames.

**Weaknesses:**

- As in the state align block, find similarity from history frames have previously proposed [1] and it seems the frame history router (channel-wise attention) is a new one proposed in this paper. However, there is no relevant ablation study with and without frame history router, so it is hard to see the improvement caused by it.
[1] Learning Trajectory-Aware Transformer for Video Super-Resolution (CVPR22)

- There is no detailed model architectures and model parameter (and runtime) comparisons with others.

**Questions:**

- Does the size of p1 and p2 affect the results?

**Limitations:**

It is surprising that the proposed method shows excellent performance in all various experiments.
But one thing may be complemented is that there is no strong guidance for temporal consistency, and I found there is a background flickering in the deraining scene in attached video.

---

> ### Author Rebuttal · Authors · 2024-08-06
>
> We thank the reviewer for finding our work important and for their insightful and constructive comments.
>
> **W1** Frame history router ablation study.
>
> In the main paper, we conducted ablation experiments to analyze the effects of different components in CHM. We investigated three setups: No CHM, No State Align Block (only Frame History Router), and Turtle (both FHR and SAB), as shown in Table 9 (in the main paper). For clarity, we have reproduced the table below. The results demonstrate that using only CHM results in a +0.23 dB increase in PSNR and adding SAB on top of FHR provides an additional +0.19 dB increase in the ablation experiment.
>
> **Methods**| **PSNR** |
> ------------------------------------------------------------|----------|
> | No CHM  | 31.84    |
> | No State Align Block (Only Frame History Router)           | 32.07    |
> | Turtle (Both State Align Block, and  Frame History Router) | **32.26**    |
>
> **Table 1**: _Ablating the CHM block to understand if both State Align Block, and Frame History Router are necessary._
>
> The paper "Learning Trajectory-Aware Transformer for Video Super-Resolution (TTVSR)" [36] processes the current frame and multiple neighboring frames together, which is inefficient. Both TTVSR and Turtle use attention to find similar patches, but Turtle has distinct advantages. Turtle uses a history state $H_t$ and reuses features from past frames, reducing computational inefficiencies. While TTVSR learns the entire trajectory of each pixel (or group of pixels) in the video, Turtle limits the history and relevant patches. This is beneficial because the entire trajectory is often not useful in video restoration due to rapid changes caused by degradations like snow, rain, and raindrops.
>
> Additionally, Turtle's Frame History Router (FHR) efficiently performs channel attention to correlate each patch in the current frame with the most relevant historical features, avoiding the overhead of full attention between frames while maintaining high restoration quality. The superiority of our proposed method is supported by the results in Table 7 and Table 5 in the main paper, which show that Turtle significantly outperforms TTVSR. Specifically, Turtle achieves a +3.96 dB increase in PSNR for Video Raindrop and Rain Streak Removal and a +1.7 dB increase in video super-resolution.
>
> **W2.** Runtime comparisons with other methods.
>
> We profile Turtle and compare it with three previous state-of-the-art video restoration methods, ShiftNet, VRT, and RVRT on varying spatial resolution sizes in terms of runtime per frame (ms), GPU memory usage, FLOPs (G), and MACs (G), refer to Table 1 in global rebuttal. Turtle can process videos at varying spatial resolutions on a single 32 GB GPU, while all three other methods throw out-of-memory errors since the memory requirements exceed the total available memory.
>
> **Q1.** Effect of patch size.
>
> CHM (specifically SAB) is effective with reasonable patch sizes. If the patch size is too small, attention computations shift to a pixel-to-pixel level, increasing the computational cost and making it harder to find similar patches since each pixel holds limited information. Conversely, if the patch size is too large, topk filtering suffers due to excessive similarities across the frame's spatial resolution. Based on our experiments, a patch size of 3x3 or 4x4 yields optimal results.
>
> **L1.** No strong temporal guidance.
>
> For the purpose of this work, we opted for a simple L1 loss function without any additional auxiliary losses. However, coupling some temporal consistency-based loss functions with the restoration loss can potentially alleviate such flickering [86] in very high frame-rate videos.
>
> [86] Dai, Peng, et al. "Video demoireing with relation-based temporal consistency." Proceedings of the IEEE/CVF conference on computer vision and pattern recognition. 2022.

---

> > ### Comment · Reviewer_PswG · 2024-08-12
> >
> > Thanks for your detailed rebuttal and my concerns have been well addressed.

---

> ### Author Response · Authors · 2024-08-13
>
> Dear Reviewer PswG,
>
> We are glad that our clarifications have addressed your concerns. Your input has been invaluable throughout the review process. We thank you for your effort in reviewing our paper and for finding it important.

---

### Official Review · Reviewer_BXMC · 2024-07-06

**Soundness:** 3
**Presentation:** 3
**Contribution:** 2
**Rating:** 5
**Confidence:** 4

**Summary:**

This paper proposes a truncated causal history model (TURTLE) for video restoration. TURTLE is in a U-Net manner with a historyless encoder and a history-based decoder. The decoder has a causal history model (CHM), which is the core part of the TURTLE. The CHM injects history frames into a hidden state $\mathbf{H}_t$ with the state align block, which is based on a cross-attention mechanism and a top-k selection strategy. The current frame is then fused with the hidden state with cross-frame attention. Experiments on several tasks demonstrate that the proposed methods can outperform the comparable methods.

**Strengths:**

1. The hidden state strategy provides a new thought for video processing tasks.
2. The proposed method outperforms comparable methods on several video restoration tasks.

**Weaknesses:**

1. The organization of ablation studies is inappropriate.
2. Some of the datasets are uncommon.
3. Some key information is lacking.

**Questions:**

1. The organization of ablation studies is inappropriate.
1.1 It's improper to put the tables of ablation studies into the appendix.
The ablations of some parts are missing.
1.2 The ablation study of top-k fusion in the state align block is missing. Most of the time, the information fusion operation in the top-k selection position of CHM is implemented by a softmax. How about using a softmax to replace top-k, i.e., remove the top-k selection strategy?
1.3 The implementation of the ''No CHM'' in the ablations is unclear. It states ''two frames are concatenated and fed to the network directly'', which is unclear. What network are the two frames fed to?
1.4 What is the patch size $p$ in the state align block? How about using different $p$?

2. There are important video restoration methods absent in the comparable tables.
[1] K. Zhou, et al., Revisiting Temporal Alignment for Video Restoration. CVPR 2022.
(Link of codes: https://github.com/redrock303/Revisiting-Temporal-Alignment-for-Video-Restoration)
[2] D. Li, et al., A Simple Baseline for Video Restoration with Grouped Spatial-Temporal Shift. CVPR 2023.
(Link of codes: https://github.com/dasongli1/Shift-Net)

3. Recurrent structures in neural networks can bring considerable costs. How does the CHM? In other words, the comparisons of running time and GPU memory cost are absent.

4. Some of the datasets are uncommon. For example, in the video super-resolution experiments, this paper selects MVSR4X as the dataset. However, the recent comparable methods (such as BasicVSR and VRT)  use REDS and Vimeo-90k as benchmarks. It seems that in the Table. 7, the TURTLE outperforms BasicVSR++ a lot, but is the BasicVSR++ model used in this paper trained for being evaluated on MVSR4X?

5. Do all the models for each task in the paper use the same number of input frames? If not, how to explain the rationality of the experiments?

6. What are the training datasets for each task? Are all the methods in the same comparable table training in the same dataset (or under the same settings)?

7. It looks like the hidden state $\mathbf{H}$ in the CHM needs some frames to start. How to handle the boundary frames of videos？

8. This paper emphasizes the proposed casual history model, but in the method part, it looks like the core part is a module for alignment (state align block). For extracting features from hidden states for restoration, what is design motivation? It would be better to introduce the components in CHM except the state align block with more details.

**Limitations:**

Please see the Questions part.

---

> ### Author Rebuttal · Authors · 2024-08-06
>
> We thank the reviewer for their time and insightful comments.
>
> **Q1.1.**
> We placed ablation studies in appendix given space limit, but can move them to the main paper.
>
> **Q1.2.**
> During rebuttal, we have added ablation experiments comparing softmax and topk (k=5), as shown in Table 2 of the one-page PDF. The main paper included experiments on the effect of k in topk patch selection in Table 11. Softmax is a special case of topk method with k equal to the total number of patches in the frame (retrieving all patches). We can see that topk is better than Softmax, as it prevents the inclusion of unrelated patches and contributes critically to performance and computation efficiency.
>
> **Q1.3.**
> Turtle takes a single frame from the video at time $t$ as input; the encoder processes this frame, while the middle and decoder stages condition restoration on historical frames. In `No CHM` setting, we concatenate frames at times $t$ and $t-1$, which are fed into the same architecture as presented in Fig. 1(a) and Sec. 3.1, without CHM blocks in middle or decoder stages—only Transformer blocks are used.  We used the same-sized models by adjusting the dimensions/blocks in the `No CHM` setting to match the size of Turtle.
>
> **Q1.4.**
> We use a patch size of 3x3. CHM is effective as long as the patch size is reasonable. If the patch size is too small, attention computations shift to a pixel level, making it harder to find similar patches since each pixel holds limited information. Conversely, if the patch size is too large, top-k filtering becomes less effective due to excessive similarities across the frame's spatial resolution. Therefore, a patch size of 3x3 or 4x4 yields optimal results.
>
> **Q2.**
> We added comparisons to RTA (CVPR'22) for video denoising at two noise levels, sigma = 30, 50, in Table 3 in one-page PDF. Unlike RTA that requires both the frame and an additional noise map as input, Turtle is completely blind to noise levels when processing a degraded input frame, making the task harder. We did not include ShiftNet as it is significantly more costly than Turtle, although runtimes and other comparisons are presented in Table 1 for a reduced ShiftNet on a single GPU in Global Rebuttal.
>
> **Q3.**
> We profile Turtle and compare it with 3 state-of-the-art video restoration methods, ShiftNet, VRT, and RVRT on varying spatial resolutions, in Table 1 of Global Rebuttal. Turtle can process videos at varying spatial resolutions on a single 32GB GPU, while others report out-of-memory errors as the resolution increases. A major benefit of Turtle is that it processes a single frame and uses the truncated history, instead of processing several frames in parallel in multiple branches, and thus is significantly faster and more efficient.
>
> **Q4.**
> Our focus is on video restoration, while we used MVSR4x to show Turtle's generalizability to more tasks. In fact, MVSR4x is a more recent real-world SR dataset from CVPR 2023, while VRT and BasicVSR were introduced earlier in 2022 and 2021 and thus didn't evaluate on MVSR4x. MVSR4x is also a challenging dataset, featuring lower resolution frames from phone cameras, resulting in lower PSNR scores compared to REDS and Vimeo90K.
>
> Yes, BasicVSR++ is also trained on the MVSR4x dataset and tested on the heldout testset of MVSR4x. We have carefully ensured all comparisons are fair. In all tables, all the methods are directly trained on the same dataset and follow the same procedure as Turtle. We either take results from their original work or the work that introduced the dataset and retrained these methods on the proposed dataset after carefully tuning the methods (the case of MVSR4x paper, which retrained BasicVSR++ when introducing the dataset).
>
> **Q5.**
> They do not. ShiftNet uses a context length of 50 frames, while VRT and RVRT utilize 16 frames. RTA uses 5 context frames. These methods compare against each other according to standard practices in video restoration literature. Although Turtle uses a truncated history of 3 frames, yet it achieves state-of-the-art results across different tasks, demonstrating its efficiency.
>
> **Q6.**
> Yes, to ensure complete fairness in comparisons, all methods we compare are trained on the same dataset and follow the same train/test splits recommended in the literature. The datasets used are listed in Appendix F of the main paper.
>
> **Q7.**
> Boundary conditions occur in the first $\tau$ frames. The first frame is restored without conditioning, second frame using the first frame as history, third frame using the first two frames and so on. Frame $\tau+1$ will use history up to the set truncation factor $\tau$.
>
> **Q8.** As Sec. 3.2 describes, overall rationale of CHM is best explained by Eqs. (1)(2), where (1) retrieves the relevant information from the prior frames $H_t$, transforms it into channels of (succinct) hidden features $\hat{H_t}$ via $\phi_t$, while (2) aims to attend current frame $F_t$ to this succinct historical feature to restore $F_t$ via $\psi_t$. In detail, Eq (1) is implemented by State Align Block (SAB) $\phi_t$, which for each patch in $F_t$, retrieves and only keeps topk relevant patches from a prior frame (in the truncated history) and “moves” those spatially to align with this patch as additional channels. This ensures that each patch in $F_t$ is not to be attending to all patches in a prior frame or to a patch at the same position in prior frame (since pixels have moved), but only to most relevant ones. Eq (2) is implemented by Frame History Router (FHR) $\psi_t$, which achieves temporal correlation across frames through efficient channel attention with succinct historical features (instead of full attention between frames). For each patch in $F_t$, FHR “routes” attention to prior historical features that are the most helpful in restoring this patch. Therefore, both (1) and (2), i.e., both SAB and FHR are core to this design and work together to achieve said feature reduction and computational efficiency of Turtle.

---

> > ### Comment · Reviewer_BXMC · 2024-08-12
> >
> > Thanks for your rebuttal. I raised my rating to borderline reject.

---

> > > ### Author Response · Authors · 2024-08-13
> > >
> > > Dear Reviewer BXMC,
> > >
> > > Thank you for your reply. We appreciate your comments and time spent on reviewing this paper again. In the rebuttal, we have addressed your main questions/concerns raised in your review, including:
> > >
> > > 1. Ablation Studies: We provided the ablation study for the proposed top-k fusion in the state align block vs. softmax as suggested, and demonstrated the effectiveness of top-k.
> > >
> > > 2. Comparisons to more baselines: We added comparisons to ShiftNet and RTA as suggested, highlighting Turtle’s advantages on performance and efficiency.
> > >
> > > 3. Efficiency/cost evaluation: We provided detailed comparisons of running time and GPU memory cost, showing Turtle’s efficiency across varying spatial resolutions.
> > >
> > > 4. Dataset for SR: We explained our use of the MVSR4X dataset as a recent and challenging benchmark for super-resolution task, as an additional task we evaluate on, and clarified our fair comparison to BasicVSR++ trained on the same dataset.
> > >
> > > 5. Extensive and fair comparisons with baselines: We ensured fairness by using the same settings, training/testing data splits, and careful tuning across all the methods evaluated. We have done extensive experimental studies and comparisons on a range of tasks and datasets.
> > >
> > > 6. CHM Design Rationale: We provided a detailed explanation of the design rationale behind the Casual History Model (CHM), and clarified all its critical components.
> > >
> > > We hope the above rebuttal has addressed and cleared all your major questions or doubts raised in the review. And we hope these responses can be thoroughly taken into account to reach into conclusions. Please let us know if you have remaining concerns/questions for this work.  Thanks again for your time!

---

> ### Author Response · Authors · 2024-08-11
> **Inquiring about additional questions/concerns.**
>
> Dear Reviewer BXMC,
>
> Thank you very much for your time spent on reviewing our work. In response to your questions, we had provided detailed explanations to clarify the points discussed and addressed the concerns you highlighted. As the deadline for the discussion period is quickly approaching, we are keen to know if our responses have addressed all your concerns. We are committed to answering any further questions that you may have.
>
> Thank you again for your valuable time and expertise.

---

> ### Comment · Reviewer_BXMC · 2024-08-13
>
> Dear authors,
> Thank you for your reply. Some of my questions are not addressed well.
> 1. Comparisons: Shiftnet is not compared. Since the re-training is in the same settings as the proposed method, why not modify the number of input frames?
> 2. Cost: There are many methods in this paper, only the costs of ShiftNet, VRT, RVRT, and Turtle are shown.

---

> ### Author Response · Authors · 2024-08-13
> **Addressing Further Questions**
>
> Dear Reviewer BXMC,
>
> Thank you for your detailed feedback and questions. We appreciate the opportunity to address your queries and provide further clarification.
>
> **1. Comparison to ShiftNet.**
>
> We had indeed trained ShiftNet with a context of 3 frames to match the input settings of the proposed method, Turtle, on the synthetic video deblurring (GoPro). However, we did not include these results initially in the manuscript because the original ShiftNet was designed for a context length of 50 frames (or even more according to their paper and repo), relying on high-end and multiple GPUs, which is a setting not straightforwardly comparable to our method and other baselines evaluated in this work. However, per the reviewer's request, we now provide these results in comparison to ShiftNet (retrained with 3-frame context on GoPro) in the following table.
>
> | **Method** | **PSNR** | **SSIM** | **MACs (G)** | **Inf. Time (ms)** |
> |-|-|-|-|-|
> |ShiftNet|33.20| 0.962|399|165|
> |Turtle|**34.50**|**0.972**|**181**|**95**|
>
> _**Table 1:**_ Comparison to ShiftNet on the video Deblurring task (GoPro) in terms of PSNR/SSIM, and profiling on input resolution of 256x256x3 on a single 32GB GPU.
>
> We can see that ShiftNet-3frame is almost 1db lower than Turtle and is not even competitive as compared to other baselines in Table 4 runnable on a single v100 GPU. The much lower performance of ShiftNet with a 3-frame context is because its original design needs to perform restoration jointly for a larger number (50 or more) of frames together to incorporate motion-compensated neighboring information.
>
> Additionally, ShiftNet's high computational demands are notable in the community. Although their paper does not specify the compute requirements, their GitHub repository indicates that training a model with a batch size of 1 and 13 frames required 8x 32GB V100 GPUs. The full 50-frame context model's compute and memory requirements at inference remain unspecified, and there have been multiple comments on the computational challenges faced when running their open-source implementation:
>
> [1] https://github.com/dasongli1/Shift-Net/issues/1 {/7,/9}
>
> Another nuance to be considered is that ShiftNet by design assumes the availability of the future frames and restores all frames in the context jointly, while the proposed method Turtle has the strength that it does not rely on future frames and can be applied in real-time scenarios (e.g., video streaming).
>
> **2. Cost Comparisons**
>
> In Table 1 of the global rebuttal, we chose to compare the compute costs, inf. time, and memory requirements of Turtle to VRT [33] (2022, published in 2024), ShiftNet [28] (2023), and RVRT [32] (2022), because they are all general video restoration techniques, which is the same as Turtle, rather than designed for accomplishing a specific task. Methods that are designed for specific restoration tasks are optimized for those tasks and, therefore, are not intended to perform competitively across a broad range of restoration tasks as Turtle does.
>
> We can see from Table 1 in global rebuttal that while methods like ShiftNet, VRT, and RVRT exhibit exponential growth in GPU memory requirements as resolutions increase, Turtle features linear scaling in GPU memory usage for higher resolutions, underscoring its computational efficiency advantage.
>
> However, in Table 8 of the main paper, we have also provided MACs comparisons to more different video restoration methods other than the ones listed in global rebuttal, including some task-specific ones.
>
> The methods for cost comparison in Table 8 are selected as follows. First, we included two general methods, RVRT and VRT, due to their high citation counts and popularity, recent publication, and competitive performance. Second, for the task-specific methods, our selection was based on their respective performances (focusing on those that ranked second or third best in respective tables), while also depending on the availability of open-source implementations (e.g., for SVDNet [9] and MetaRain [46], code is not available, which are also worse than Turtle on task performance). Furthermore, if the performance gap between a method and Turtle was larger than 1dB PSNR (which is a substantially large gap in log scale), we would not consider those methods for cost comparison, since their task performance is not even close.
>
> Table 8 and the extensive task results presented in Tables 1-7 suggest that Turtle either substantially outperforms the baselines or achieves the leading results on all the major video restoration tasks evaluated while still being more computationally efficient. For example, EDVR and BasicVSR, RDDNet have relatively lower GMACs in Table 8, although still higher than Turtle, but they achieve much lower performance than Turtle in Table 7 and Table 5. These results have verified the effectiveness and efficiency of Turtle as compared to baselines.
>
> Thank you again, and we hope these clarifications address your concerns.

---

> > ### Comment · Reviewer_BXMC · 2024-08-14
> >
> > Dear authors,
> > Thank you for your clarification, I've raised my rating to be positive.

---

> ### Author Response · Authors · 2024-08-14
>
> Dear Reviewer BXMC,
>
> Thank you for your valuable input, insight, and feedback that helped improve our work. We appreciate your positive stance of our work, and your decision of raising the score.

---

### Official Review · Reviewer_SsXx · 2024-07-11

**Soundness:** 3
**Presentation:** 2
**Contribution:** 3
**Rating:** 5
**Confidence:** 5

**Summary:**

This work presents a video restoration framework named TURTLE, which stands for truncated casual history model. The key innovation of TURTLE is its ability to efficiently model the transition dynamics of video frames governed by motion, a critical challenge in video restoration. Unlike traditional methods that process multiple contextual frames simultaneously, TURTLE enhances efficiency by storing and summarizing a truncated history of the input frame's latent representation into an evolving historical state.

**Strengths:**

1.	The paper introduces a novel video restoration technique that leverages a truncated causal history model, which is a unique approach to handling the transition dynamics of video frames influenced by motion. This represents a significant advancement in the field of video processing.
2.	The TURTLE framework achieves state-of-the-art results across a wide range of video restoration tasks, including desnowing, deraining, raindrop and rain streak removal, super-resolution, deblurring, and blind video denoising. The consistent high performance across different tasks underscores the effectiveness of the proposed method.
3.	TURTLE is designed to be computationally efficient, which is a critical consideration for practical applications. The framework reduces computational costs compared to existing methods while maintaining high performance, making it suitable for resource-constrained environments.

**Weaknesses:**

1. The section on related work is seriously lacking in content; the author should revise this section to include a more detailed explanation of related work, especially in the areas of temporal modeling and Causal Learning.

2. There is a scarcity of Visual Comparisons, and many tasks lack real-world sample comparisons. This is a serious issue, and I hope the author can provide a complete set of comparisons.

3. Why is it necessary to use a Casual History Model in each decoder? I believe applying CHM in the latent space should be sufficient. This raises concerns about the actual inference efficiency of the model.

4. There is a lack of comparison regarding actual inference time, as well as a crucial study on the actual VRAM usage at high resolutions, both of which are very important; the reference value of GMacs alone is limited.

**Questions:**

Why is it necessary to use a Casual History Model in each decoder? I believe applying CHM in the latent space should be sufficient. This raises concerns about the actual inference efficiency of the model.
There is a lack of comparison regarding actual inference time, as well as a crucial study on the actual VRAM usage at high resolutions, both of which are very important; the reference value of GMacs alone is limited.

**Limitations:**

Please refer to the weaknesses.

---

> ### Author Rebuttal · Authors · 2024-08-06
>
> We thank the reviewer for their valuable feedback and reading our work.
>
> **W1.** The section on related work is seriously lacking in content.
>
> Due to space constraints, we had to be selective in related work. In the rebuttal, here we would like to add a literature review on temporal modeling and causal learning for videos as suggested.
>
> **Temporal Modelling:** In video restoration, temporal modeling mainly focuses on how the neighboring frames (either in history or in the future) can be utilized to better restore the current frame.
> For such a procedure, the first step usually involves compensating for motion either through explicit methods (such as using optical flow [32], [33], [35], [5]), or implicitly (such as deformable convolutions [61], search approaches [62], or temporal shift [28]). A few works in the literature focus on reasoning at the trajectory level (i.e., considering the entire frame sequence of a video) [36] through learning to form trajectories of each pixel (or some group of pixels). The motivation is that in this case, each pixel can borrow information from the entire trajectory instead of focusing on a limited context. The second step is then aggregating such information, where in the case of Transformers, self-attention is employed, while MLPs are also used in other cases.
>
> **Causal Learning for Videos:** In videos, causal learning is generally explored in the context of self-supervised learning to learn representations from long-context videos with downstream applications to various video tasks such as action recognition, activity understanding, etc. In [85], causal masking of several frames at various spatio-temporal regions as a strategy to learn the representations is explored. To the best of our knowledge, almost all of the state-of-the-art video restoration methods are not causal by design since they rely on forward and backward feature propagation (i.e., they consider both frames in history and in the future) either aligned with the optical flow or otherwise [32], [33], [5].
>
> [85] Bardes, Adrien, et al. "Revisiting feature prediction for learning visual representations from video." arXiv preprint arXiv:2404.08471 (2024).
>
> **W2.** Visual Comparisons.
>
> We have already provided extensive visual comparisons between Turtle and state-of-the-art baseline methods in the main paper for a range of tasks on respective benchmark datasets, including desnowing in Figures 3 and 11, night deraining in Figure 3, deblurring in Figures 4 and 9, raindrop removal in Figure 4, and blind video denoising in Figure 5. These visual comparisons on done on the benchmark dataset where the methods are evaluated in order to align with the reported numerical performance comparisons in tables, which is following standard practice in the literature. We also have included a visual comparison on real-world video super-resolution in Figure 5.
>
> During the rebuttal, we added more real-world Visual Effect results, including real-world deblurring from the BSD 3ms-24ms dataset and removing snow from real snowy videos taken from www.pexels.com (a free stock videos website). Please refer to Figure 1 in the one-page PDF for the visual results.
>
> These results when put together show the competence of Turtle not only numerically but also in terms of visual effects.
>
> **W3.** Is CHM in Latent sufficient?
>
> In rebuttal, we add an ablation study for the potential benefits of adding CHM at the decoder stage compared to adding it only at the latent stage. The ablation results are included in Table 1 of the one-page PDF. This table is also shown below:
>
> | **Ablations**                    | **PSNR** |
> |----------------------------------|----------|
> | No CHM                           | 31.84    |
> | CHM in Latent                    | 32.05    |
> | CHM in Latent & Decoder (Turtle) | **32.26**    |
>
> **Table 1**: _Ablation experiments on CHM placement in latent, and latent and decoder._
>
> Our experiment indicates that having CHM in both the latent and decoder stages is necessary for optimal performance. In the latent stage, the spatial resolution is minimal, and CHM provides greater benefit in the following decoder stages as the spatial resolution increases.
> Furthermore, we also calculated the computational overhead of CHM, and show that out of 181 MACs (G) of Turtle, all CHM blocks collectively contribute 11.8 MACs (G), which accounts for 7% of the overall computation cost to achieve temporal modeling. In contrast, processing and restoring multiple frames in parallel with ShiftNet [28], learning trajectories in TTVSR [26], or deploying an additional optical flow network [32,33] entail considerably higher costs.
>
> **W4.** Lack of comparison at different resolutions.
>
> We profiled the proposed method Turtle and compared it with previous video methods ShiftNet, VRT, and RVRT. We compute per-frame inference time (ms), MACs (G), FLOPs (G), and GPU memory usage (MBs) at varying input resolutions (refer to Table 1 in global rebuttal).
>
> In practice, note that ShiftNet considers a context of 50 frames, VRT considers a context of 16 frames, and RVRT considers a context of 16 frames for denoising and deblurring. For superresolution, a total of 30 frames are fed. Note that both VRT and RVRT also rely on an optical flow architecture (SpyNet) and fine-tune it during training for the restoration procedure. These factors significantly increase their computational costs and limit their deployability.
>
> However, Turtle stands out by only considering a single frame and conditioning the restoration on the history of the current frame up to a truncation factor (τ =3). This setup significantly enhances efficiency, allowing Turtle to perform inference at varying spatial resolutions on a single GPU. This capability is crucial for the successful deployment of video restoration models on hardware-constrained devices, where the availability of multiple GPUs is often limited or impractical.

---

> ### Author Response · Authors · 2024-08-11
> **Follow up and Inquiry for Further Questions/Concerns.**
>
> Dear Reviewer SsXx,
>
> Thank you very much for your time, and insightful comments. We had provided detailed explanations to clarify the points, and questions raised. As the deadline for the discussion period is quickly approaching, we were just wondering if all of your questions have been addressed. We are committed to answering any questions/concerns that you may have.
>
> Thank you again for your valuable time.

---

### Official Review · Reviewer_KLsv · 2024-07-11

**Soundness:** 3
**Presentation:** 2
**Contribution:** 4
**Rating:** 6
**Confidence:** 4

**Summary:**

The paper presents a novel framework TURTLE for video restoration. TURTLE aims to improve video restoration tasks by modeling and utilizing truncated historical data of input video frames to enhance the restoration quality while maintaining computational efficiency. The proposed method demonstrates state-of-the-art results across various video restoration benchmarks, including desnowing, deraining, deblurring, and super-resolution.

**Strengths:**

- The introduction of a truncated causal history model is novel and addresses the limitations of both parallel and recurrent video processing methods.
- The paper thoroughly explains the TURTLE architecture and its components, including the encoder, decoder, and Causal History Model (CHM).
- The method achieves superior results on multiple video restoration benchmarks, showing clear improvements over existing techniques. The method is evaluated across various video restoration tasks quantitatively and qualitatively.

**Weaknesses:**

- This paper did not discuss the parameters and FLOPs of this model.
- Lack of comparison with  Restormer: Efficient transformer for high-resolution image restoration. (CVPR2022), A Simple Baseline for Video Restoration with Grouped Spatial-temporal Shift.  (CVPR2023)
- Lack of visualization results of the effectiveness of truncation factor and value of topk.

**Questions:**

-  Line 24, the author claims recurrent designs often result in cumulative errors. I think it would be better to explain why a truncated causal history model can avoid this phenomenon.
- As shown in Fig. 7, how many frames are affected by incorrect tracking query points? Will this reduce the restoration quality of subsequent frames?
- I noticed a latent cache block in your code that is barely mentioned in the paper. Could you explain its function and impact on the model's performance?
- Please refer to the weaknesses above.

**Limitations:**

The authors have largely addressed their limitations.

---

> ### Author Rebuttal · Authors · 2024-08-06
>
> We thank the reviewer for reviewing our work and providing constructive comments.
>
> **W1.** FLOPs of Turtle.
>
> We have provided the Turtle's MACs (G) in Table 8 of the main paper and discussed its comparison to several previously published video restoration methods in Section 4.8. Additionally, during rebuttal, we have profiled the proposed method, Turtle, and compared it with previous methods, including ShiftNet [28], VRT [32], and RVRT [33], as reported in Table 1 of the global rebuttal. We computed per-frame inference time (ms), MACs (G), FLOPs (G), and GPU memory usage (MB).
>
> From these comparisons, we can see Turtle is significantly more computationally efficient. ShiftNet considers a context of 50 frames, VRT considers 16 frames, and RVRT also uses 16 frames. Both VRT and RVRT rely on fine-tuning an optical flow model (SpyNet) to restore a video. In contrast, Turtle only considers a single frame and conditions its restoration on the recent history of the current frame up to a truncation factor (τ = 3), which reduces computational costs and enhances efficiency.
>
> This computational efficiency is achieved by the proposed CHM which contains a state align block (SAB) to retrieve only top-k relevant patches from recent frames to help restore each patch, and a Frame History Router (FHR) to further route attention to features in the most relevant historical frame. Both SAB and FHR together achieve efficient temporal cross-frame attention calculation.
>
>
> **W2.** Lack of comparison with Restormer, and ShiftNet.
>
> Restormer [76] is designed for Image Restoration and does not account for the temporal nature of video frames. Despite this difference, here we provide a comparison with Restormer on the GoPro video deblurring task. Unlike video restoration methods, Restormer treats video frames as individual images. Consequently, it is uncommon in the literature to compare video restoration methods with image restoration methods as it is not a fair comparison.
>
> | **Methods** | **Task**          | **PSNR** | **SSIM** |
> |-------------|-------------------|----------|----------|
> | Restormer   | Image Restoration | 32.92    | 0.961    |
> | Turtle      | Video Restoration | **34.50**    | **0.972**    |
>
> **Table 2**: _Comparison of Turtle with Restormer on the GoPro deblurring task in terms of PSNR and SSIM metrics._
>
> We have not compared our model with ShiftNet due to the immense difference in computational cost, where a major benefit of Turtle is its low computation cost. In Table 1 (refer to the global rebuttal), we provide a detailed comparison of Turtle with ShiftNet [28] in terms of FLOPs, GPU memory consumption, and inference time for a single frame on a 32 GB GPU. To compute the numbers in Table 1, we considered an 8-frame context for ShiftNet, as it was not feasible to run even the smallest ShiftNet model with its full 50-frame context length, which was used to report the performance in their paper, on a single 32 GB GPU. Additionally, it is important to note that ShiftNet considers a 50-frame context, both in the future and in history. Therefore, unlike Turtle, ShiftNet is not suitable for online video restoration (a typical use case in streaming scenarios) since it relies on future frames.
>
>
> **W3.** Lack of visualization results of the effectiveness of truncation factor and value of top-k.
>
> During rebuttal, we have added the visualization results for the value of top-k (with k = 5 and k = 20); refer to Fig. 2 in the one-page PDF. As for the effect of the truncation factor, increasing the truncation factor will increase computational costs. Our experiments, presented in Table 10, revealed that raising the truncation factor from 3 to 5 did not enhance PSNR scores or visual quality and merely increased computational expenses. Conversely, increasing the truncation factor from 1 to 3 led to visual improvements and higher PSNR scores, which justifies the slight rise in computational cost. Thus, we have chosen a truncation factor of 3 to be used in Turtle.
>
> **Q1.** How truncated causal history model can avoid cumulative errors?
>
> This can be seen through Eq (1) and (2). Although like RNN models, in Turtle, the causal history state $\hat{H_t}$ given by (1) will also recursively depend on prior causal history state $\hat{H_{t-1}}$, yet we notice through Eq(1) and (2) that when decoding $y_t$, Turtle only needs to rely on the current frame $F_t$ and the recent truncated history of $H_t$, where $\hat{H_t}$ (although recursive itself) is extracted by retrieving topk patches from $H_t$ to align with each patch in $F_t$ as additional channels. Therefore, the errors only depend on $F_t$ and $H_t$ with a small truncation factor and do not accumulate.
>
> **Q2.** As shown in Fig. 7, how many frames are affected by incorrect tracking query points? Will this reduce the restoration quality of subsequent frames?
>
> In some cases, where the input frames are extremely degraded or occluded (for instance, with snow covering the point of interest), the points found in the previous frames can be slightly wrong. In Fig. 7, we see this phenomenon with Zebra's front part of the torso. In frames at time t-1, t-2, etc., the region of interest is blurred and occluded due to snow and haze. Therefore, in such cases, some mismatch can occur. However, notice that the most similar points are still somewhere on the Zebra's torso, and not on other random regions (like grass).
>
> **Q3.** Latent Cache Block in code.
>
> The latent block (or latent cache block as in the code, or Middle Block in Figure 1 in the main paper) is no different than the decoder blocks in principle. The only difference is in the number of CHM blocks.

---

> ### Author Response · Authors · 2024-08-11
> **Please let us know if you have further questions/concerns**
>
> Dear Reviewer KLsv,
>
> Thank you very much for your time spent on reviewing our work.
> We had provided detailed explanations to clarify the points raised. As the deadline for the discussion period is quickly approaching, we are just wondering if all your questions have been addressed. We are committed to address any questions/concerns that you may have.
>
> Thank you again for your valuable time.

---

> > ### Comment · Reviewer_KLsv · 2024-08-13
> >
> > Thank you for your rebuttal. Most of my concerns have been addressed. I've raised my rating.
> >
> > Best regards.

---

> ### Author Response · Authors · 2024-08-13
>
> Dear Reviewer KLsv,
>
> We thank you for your decision to increase the score and your effort in reviewing our paper. We appreciate your acknowledgement that our rebuttal has addressed your concerns. Your insights have helped improve our paper.

---

### Author Rebuttal · Authors · 2024-08-06

## **Profiling Turtle**

We profile the proposed method, Turtle, in terms of per-frame inference time (in ms), MACs (G), FLOPs (G), and GPU memory usage (in MBs) on a single 32GB Nvidia V100 GPU. ShiftNet uses a context length of 50 frames and restores all 50 frames together, while VRT uses (a context length of) 16 frames, RVRT uses 16 frames for deblurring and denoising, and 30 frames for super-resolution. However, these settings are not runnable on a single V100 GPU, which reports out of memory (OOM). Thus, for the purpose of generating this table, we chose a context length of 8 frames for ShiftNet and 2 frames for RVRT and VRT.

|  Methods |  Resolution | Per Frame Inference Time (ms) | MACs (G) | FLOPs (G) | GPU Memory Usage (MBs) |
|:--------:|:-----------:|:------------------------:|:--------:|:---------:|:----------------------:|
| ShiftNet |  256x256x3  |            190           |    989   |    1978   |          2752          |
|          |  640x480x3  |            510           |   5630   |   11260   |          7068          |
|          |  1280x720x3 |            OOM           |    OOM   |    OOM    |           OOM          |
|          | 1920x1080x3 |            OOM           |    OOM   |    OOM    |           OOM          |
|    VRT   |  256x256x3  |            455           |   1631   |    3262   |          3546          |
|          |  640x480x3  |           2090           |   7648   |   15296   |          11964         |
|          |  1280x720x3 |            OOM           |    OOM   |    OOM    |           OOM          |
|          | 1920x1080x3 |            OOM           |    OOM   |    OOM    |           OOM          |
|   RVRT   |  256x256x3  |            252           |   1182   |    2364   |          5480          |
|          |  640x480x3  |           1240           |   5294   |   10588   |          21456         |
|          |  1280x720x3 |            OOM           |    OOM   |    OOM    |           OOM          |
|          | 1920x1080x3 |            OOM           |    OOM   |    OOM    |           OOM          |
|  Turtle  |  256x256x3  |            95            |    181   |    362    |          2004          |
|          |  640x480x3  |            380           |    812   |    1624   |          4826          |
|          |  1280x720x3 |           1180           |   2490   |    4980   |          11994         |
|          | 1920x1080x3 |           2690           |   5527   |   11054   |          24938         |

**Table 1**: _We profile the proposed method, Turtle, on a single 32 GB V100 GPU, and compare with 3 recent video restoration methods, namely ShiftNet [28], VRT [33], and RVRT [32]. We consider different input resolutions and compute the per-frame inference time (ms), total MACs (G), FLOPs (G), and the GPU memory usage of the model. OOM denotes Out-Of-Memory error i.e., the memory requirement exceeded the total available memory of 32GB._

**Note on References in Reviewer-specific Rebuttals**

We use same the reference numbering as in the main paper. For example, ShiftNet is the reference [28] in the main paper. For papers that we cite additionally, we provide the references in their specific replies.

---

### Decision · Program_Chairs · 2024-09-25

**Decision:**

Accept (poster)

**Comment:**

The reviewers were unanimous in accepting this paper, and rated it with an Accept, a Weak Accept, and 2 Borderline Accept.
They found the paper novel, its performance and evaluation convincing and the design computationally efficient. They raised
some concerns but many of these were addressed in the rebuttal and in the direct messages to the reviewers, and reviewers
have correspondingly adjusted their ratings.
The authors are encouraged to integrate the answers to these concerns in their paper, especially regarding the paper presentation.
The AC has read all the reviews, the rebuttal and the messages exchanged between the reviewers and the authors
and agrees with the reviewers that this paper should be accepted.